# Technical note: Extending sea level time series for extremes analysis with statistical methods and neighbouring station data

Kévin Dubois[1,2], Morten Andreas Dahl Larsen[3], Martin Drews[4], Erik Nilsson[1,2], Anna Rutgersson[1,2]

[1]Department of Earth Sciences, Uppsala University, Uppsala, 752 36, Sweden
[2]Centre of Natural Hazards and Disaster Science (CNDS), Uppsala University, Uppsala, 752 36, Sweden
[3]Danish Meteorological Institute, Copenhagen, 2100, Denmark
[4]Department of Technology, Management and Economics, Technical University of Denmark, Lyngby, 2800, Denmark

*Correspondence to*: Kévin Dubois (kevin.dubois@geo.uu.se)

**Abstract.**

Extreme sea levels may cause damage and disruption of activities in coastal areas. Thus, predicting extreme sea levels is essential for coastal management. Statistical inference of robust return level estimates critically depends on the length and quality of the observed time series. Here we compare two different methods for extending a very short (~10 years) time series of tide gauge measurements using a longer time series from a neighbouring tide gauge: Linear Regression and Random Forest machine learning. Both methods are applied to stations located in the Kattegat basin between Denmark and Sweden. Reasonable results are obtained using both techniques with the machine learning method providing a better reconstruction of the observed extremes. Generating a set of stochastic time series reflecting uncertainty estimates from the machine learning model and subsequently estimating the corresponding return levels using extreme value theory, the spread of the return levels is found to agree with results derived from more physically-based methods.

## 1 Introduction

Extreme sea levels (ESLs) can have disastrous consequences in the coastal zone in terms of flooding vulnerable assets, loss of lives, and disturbances (Brown et al., 2018; Vousdoukas et al., 2020; Wahl et al., 2017). Coastal floods generally result from a combination of ESL, wind, waves, tides and local conditions, including bathymetry and terrain features. Climate change also affects ESL events due to sea level rise and changes in storm frequency and/or intensity particularly (Rutgersson et al., 2021). Reliable estimates of current and future ESLs are urgently needed to mitigate the impacts of disaster risks and to inform adaptation to climate change. Long time series of observed sea levels are essential for improving confidence in statistically inferred return levels (RLs) (Menéndez et al., 2010; Woodworth et al., 2011) and are often considered essential for coastal planning. International initiatives such as the Global Sea Level Observing System (GLOSS) (Caldwell et al., 2012; Merrifield et al., 2012) and other works (Woodworth et al., 2010) have highlighted this necessity and called for recovering historical

records in what is called "data archaeology". Still, the temporal paucity of sea level time series (Holgate et al., 2013) remains a limitation for adequately estimating RLs and ESLs in many places.

This technical note evaluates a machine learning method called Random Forests (RF) (Breiman, 2001) for extending the sea level time series obtained by a tide gauge of interest using a longer time series at a neighbouring tide gauge in the context of
analysing sea level extremes. This is particularly relevant when the initial time series is very short e.g., in the order of ~10 to 20 years, which is principally too short to allow reliable statistical inference of ESLs e.g., RL corresponding to a 100-year event. The RF methodology is compared to a linear regression (LR) model, which for short time series could also be expected to perform adequately. Our study area lies within the Kattegat basin, located on the west coast of Sweden around the city of Halmstad (Fig. 1). Here the highest recorded Swedish sea level of 235 cm was observed in November 2015 according to the
Swedish Meteorological and Hydrological Institute (SMHI), this event was mainly due to local conditions leading to a sea level increase of 50 to 100 cm in comparison with neighbouring stations as Viken, the second one is a seiche effect which could have added around 25 cm to the total sea level (Johansson, 2018). In this area, tides vary with an amplitude of around 20 cm during spring tides (Svansson, 1975), and current ESLs are mainly due to storm surge effects. But also, other factors could play a role, such as the preconditioning of the Baltic Sea (Andrée et al., 2022). Hieronymus et al. (2020) have shown
that the Swedish West Coast is expected to be one of Sweden's most exposed areas due to rising sea levels.

Different methods have been proposed for extending sea level records, such as Bernier et al. (2007), who use short observation time series associated with a 40-year hindcast surge model. Reconstructions by Cid et al. (2018) are based on tide gauge data and atmospheric conditions. Hieronymus et al. (2019) show a good performance of neural networks in predicting sea levels at
tide gauges located along the Swedish coast based on different atmospheric variables and tide gauge records. Granata et al. (2021) find similar results when forecasting tides in the Venice region using different machine learning methods: Random Forest, Regression Tree and Multilayer Perceptron. Recently, Bellinghausen et al. (2023) have demonstrated the utility of using a Random Forest Classifier to satisfactorily predict the occurrence of ESLs at a few stations around the Baltic Sea within three days based on surface wind and pressure fields, precipitation and the prefilling state of the Baltic Sea.

In the following, we systematically evaluate the performance of RF as means of extending a very short time series of only ten years, reconstructing past sea level variations based on a more extended time series from a neighbouring station. This approach is compared to the linear regression approach. Both methods have previously been found to reduce biases efficiently and are relatively computationally inexpensive with low complexity, when applied to a small number of input variables as is the case
in this study. To evaluate the sensitivity of the reconstructed sea level with respect to the geographic distance to neighbouring stations, we apply the method to data from different stations. Finally, we consider the method's potential and limitations with respect to the reconstruction of sea level extremes, when the time series of interest is very short, and inherently provide a poor sampling of even moderately extreme events.

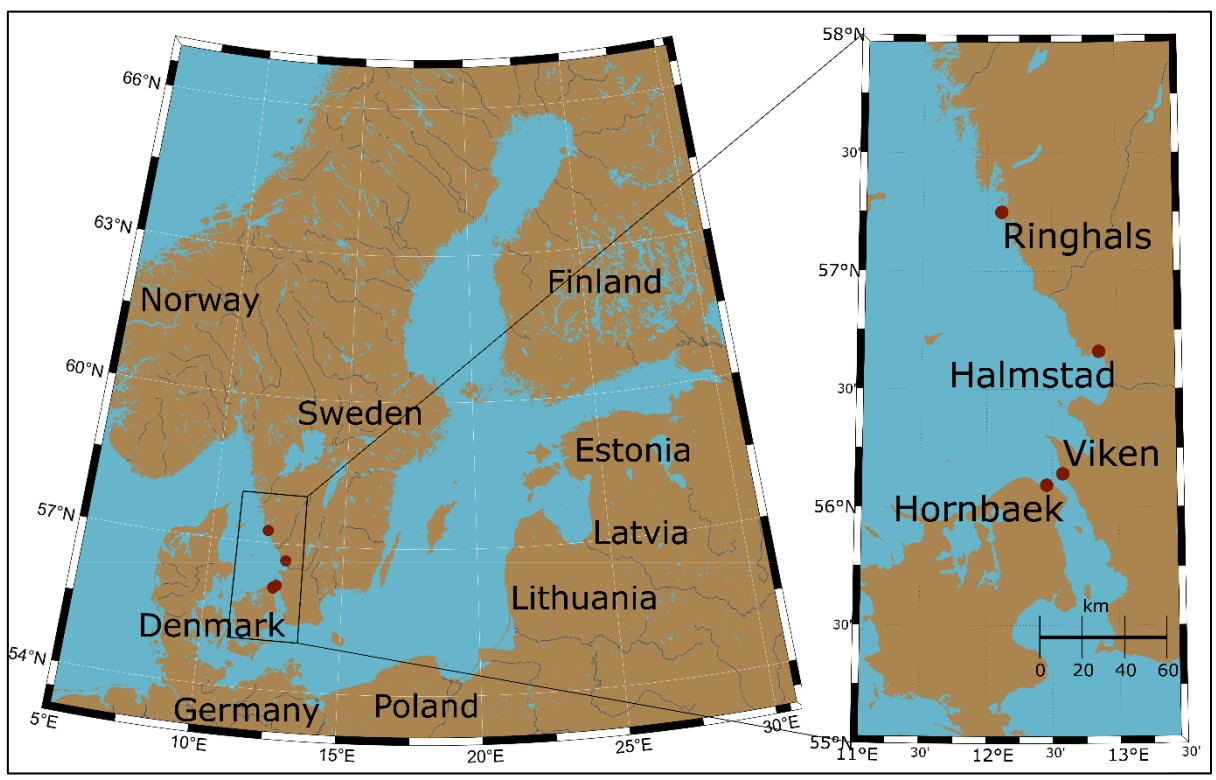

**Figure 1: Map of Northern Europe with indication of the study area and the tide gauge stations (red dots).**

## 2 Data and Methods

### 2.1 Sea level data

The datasets used in the analysis are hourly sea level observations from different stations available from SMHI (SMHI, 2022) and the Danish Meteorological Institute (DMI). Three stations are located on the west coast of Sweden: Ringhals (station number: 2105 "RINGHALS"), Halmstad (station number: 35115 "HALMSTAD SJÖV") and Viken (station number: 2228 "VIKEN"), and one station is located on the east coast of Denmark: Hornbaek (Hansen, 2007) (Fig. 1). The distance between Hornbaek and Viken is around 9 km, around 130 km between Hornbaek and Ringhals and 127 km between Viken and Ringhals (Table 1). The geographical location of the stations is important as it can change how the water level behaves, for example, if the stations are constricted in a channel as for Viken and Hornbaek. Here, ESL are defined as the total highest measured sea level including tides and storm surges, this choice is motivated because of the low tidal range in the area (Svansson, 1975).

Each hourly time series is first linearly detrended and transformed into a time series of daily maxima from which the annual maximum is determined for each year in the series. When determining the annual daily maximum, we enforce a minimum

temporal separation of 2 days to assert the independence of events at each station. The data sets are of varying lengths (Fig. 2) ranging from twelve years (Halmstad station) to 129 years (Hornbaek station). Long-term linear trends (i.e., sea level rise) were estimated over the whole time series for all stations and found to range between 0.33 cm (Ringhals), 0.35 cm (Hornbaek), 1.47 cm (Viken) and 5.51 cm (Halmstad) per decade.

After being detrended, Hornbaek sea level varies from -145 to 187 cm, Viken sea level between -114 and 166 cm, Ringhals

sea level is comprised between -105 and 162 cm and Halmstad sea level between -94 and 213 cm relative to mean sea level (Fig. 2).

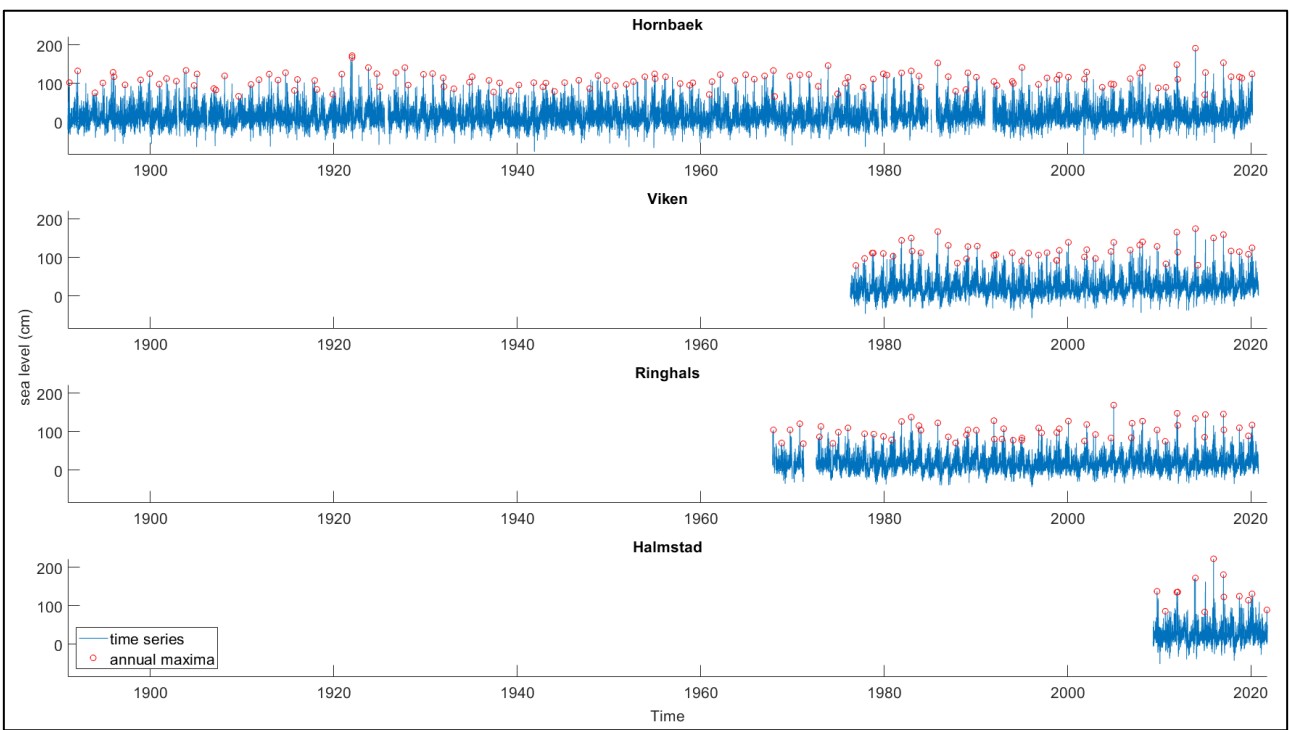

**Figure 2: Sea level time series from the four tide gauge stations, i.e., daily maximum values (blue) and yearly maximum values (red**
**circles).**

## 2.2 Methods

The proposed approach for temporally extending short observed sea level time series at the station of interest (y) is based on using longer observed sea level time series at a neighbouring station (x)constituting the predictor. For each x/y station pair (Table 1), we use a temporally coinciding time series to set up each prediction model, for example 10 years. We will denote

this the setup period. Within the setup period, models are fitted or trained using both time series for 80 % (eight years in our example) of overlapping data (training dataset), whereas the last 20 % (two years in our example) serve as validation dataset

to provide an unbiased evaluation of the model fit. The coinciding period outside the setup period constitutes the testing period. Two overall predictor approaches are employed – one using simple LR and one based on RF machine learning.

For both approaches, we only use sea level of daily maxima at station y, time t denoted $y_t$ is predicted from the sea level at station, time t denoted $x_t$. Nevertheless, since the sea level is sensitive to meteorological conditions, which are advected by the winds, the sea level at one station at time t might be better predicted if using time-lagged sea-level from another station, e.g. the sea-level at times t± a few days. Here, we are using daily maxima which might buffer this effect to some extent. However, it is most likely some slight improvements could be found when applying some time-delayed variables. Therefore, a short analysis has been done in testing time lagged variables for the setup period of 30 years for each x/y stations pair. 3 different tests have been done, the first one is the one used in this paper where no time-delayed predictors have been added: $y_t = RF(x_t)$; in the second one we add 2 times delayed variables time t-1day and time t-2days: $y_t = RF(x_t , x_{t-2}, x_{t-1})$; in the third one we add 4 times delayed variables time t-1day, time t-2days, time t+1day and time t+2days: $y_t = RF(x_t , x_{t-2}, x_{t-1}, x_{t+1}, x_{t+2})$. Slight improvements of RMSE values of around 1 to 2.5 cm as well as r values of around 0.03 to 0.08 for all x/y stations pair with best test being the third one and the second one presenting intermediate improvement. Bias values barely changed with changes of maximum 0.2 cm however, towards the extremes, values from the tests 2 and 3 present a bigger underestimation than the ones from the test 1. We then think that, for this study, the method used within the paper (test 1) is sufficient and even might be the best one to reproduce ESLs. This might be explained as the RF is only statistically based and applied on stations not so far away from each other (max 130km) which are therefore most of the time submitted to the same synoptic atmospheric phenomena within one day. Some more testing could be done to really assess the potential added values of using time delayed variables but this is outside of the scope of this study.

### 2.2.1 Linear Regression

Based on each x/y predictor-reconstruction station pair, a linear equation is found using the least squares method as means of determining the best fit coefficients. Based on the resulting equation, $y_t$ is predicted from $x_t$.

All coefficients values from the linear fits are positive and fairly close to one (0.765 to 1.12) meaning low sea level at one station corresponds to low sea level at another station and similar effect is then found for high and intermediate sea level. Therefore, the sea level at one station varies at a rather similar rate to the other one as a coefficient value of 1 would mean that the sea level measured at one station would be increasing or decreasing as the same rate at another one. The set xHornbaek/y_Halmstad presents the closer to one coefficient highlighting a strong correlation between those 2 stations. Only the xRinghals/y_Halmstad and x_Viken/y_Halmstad present a coefficient higher than 1. This suggests that the sea level at Halmstad varies at a higher pace than at the two predictor's stations.

### 2.2.2 Random Forest

A probabilistic RF model is trained using the sea level at one station as the predictor (x) and the sea level at another station as the predictand (y). The RF method yields a mean and a standard deviation for each predicted value (Breiman, 2001). The RF

model is implemented using the MATLAB function TreeBagger (https://se.mathworks.com/help/stats/treebagger.html) where the regression method is based on a number of trees and minimum leaf size hyper-parameters. The mean and standard deviation values are predicted using the MATLAB function predict (https://se.mathworks.com/help/stats/treebagger.predict.html). We use the predict function for our regression problem and in the documentation, there is further description for the function for the weighted average of the prediction using selected trees. We do not use the option of TreeWeights but do use the output of the standard deviations of the computed responses over the ensemble of the grown trees, hence for regression: [Yfit,stdevs] = predict(B,X). Here, Yfit is a vector of predicted responses for the predictor data in the table or matrix X, based on the ensemble of bagged decision trees B. By default, predict takes a democratic (nonweighted) average vote from all trees in the ensemble. These parameters are here set to 500 and 1, respectively which have been chosen after a brief sensitivity analysis and are not the default choices for regression models.

The LR is fitted and the RF model is trained using the same setup period for each station pair (Table 1).

### 2.2.3 Model Testing

To evaluate the proposed methodology, different analyses with different combinations of stations are used to test the spatial and temporal sensitivity (Table 1). Six analyses using different combinations of station data obtained at Hornbaek, Viken and Ringhals are carried out using the recent 10 full years (2010-2020) as the common setup period for model training and validation (cf. section 2.2.2). Six additional analyses are carried out, where we predict Viken sea levels from Hornbaek data using the two previous time periods (2000-2010) and (1990-2000) as well as using a 20-year setup period (1990-2010 and 2000-2020) and a 30-year setup period (1990-2020) for training and validation to evaluate the temporal sensitivity. All the 36 combinations possible have then been analysed to better estimate the spatial and temporal sensitivity. Finally, we compare reconstructed sea levels at Halmstad using the station data from Hornbaek, Viken and Ringhals, respectively (cf. section 3.2) for the period (2010-2020). In the latter case, we also estimate RLs based on the reconstructed time series and compare them to previous results reported for Halmstad.

To assess the performance of each model, different goodness-of-fit metrics (GOFs) are chosen i.e., the root mean square error (RMSE) and the Pearson correlation coefficient (*r*). Also, the general bias (*bias*) and the 95th percentile bias (*perc95-bias*) between the observations and both model reconstructions (LR and RF) within the validation period are calculated.

To evaluate the model's performance towards the extremes, annual maxima and values above the 95th, 97th and 99th percentiles from observations are compared with the corresponding predicted values.

**Table 1: Experimental setup and summary of analyses.**

| Predictor | Predictand | Setup period | | | Coinciding period | Distance between stations | Study |
|---|---|---|---|---|---|---|---|
| station x | station y | 1 | 2 | 3 | | (km) | |
| Viken | Hornbaek | 2010-2020 | | | 1977-2020 | 9 | Spatial correlation analysis |
| Ringhals | Viken | 2010-2020 | | | 1977-2020 | 127 | |
| Hornbaek | Ringhals | 2010-2020 | | | 1968-2020 | 130 | |
| Viken | Ringhals | 2010-2020 | | | 1977-2020 | 127 | |
| Ringhals | Hornbaek | 2010-2020 | | | 1968-2020 | 130 | |
| Hornbaek | Viken | 2010-2020 | | | 1977-2020 | 9 | Temporal correlation analysis |
| | | | 2000-2010 | | 1977-2020 | | |
| | | | | 1990-2000 | 1977-2020 | | |
| | | 2000 - 2020 | | | 1977-2020 | | |
| | | | 1990 - 2010 | | 1977-2020 | | |
| | | 1990 - 2020 | | | 1977-2020 | | |
| *Hornbaek* | *Halmstad* | *2010-2020* | | | *2010-2020* | *68* | *case study* |
| *Viken* | | *2010-2020* | | | *2010-2020* | *60* | |
| *Ringhals* | | *2010-2020* | | | *2010-2020* | *80* | |

## 2.2.4 RF method with random sampling to evaluate return levels (RLs)

The RF method estimates the standard deviation associated with the predicted sea level daily maximum at each time point. We denote "RF method with random sampling" the following introduced methodology. Based on the RF daily means and standard deviations, we select the corresponding annual maxima from the reproduced time series and their associated standard deviations. We assume that a Gaussian distribution describes the probability for each predicted annual maximum. RLs are subsequently calculated using a generalised extreme value (GEV) distribution fitted to the annual maxima (Coles, 2001). This

yields an ensemble of randomly drawn RL curves. The 95[th] percentile of the ensemble spread is calculated.

We denote x, the predictor time series of daily maxima; y, the predicted time series of daily maxima and $std_y$, the standard deviation associated with y so as: at time t, $(y_t, std_{y_t}) = RF(x_t)$ with RF, the trained RF model.

We can then extract the time series of annual maxima from the mean predictions and its associated standard deviation that we denote $Y_n$ and $std_{Y_n}$ with $n \in [1, N]$ with N, the number of years in y. Let's introduce a random variable R that is distributed normally such as:

$$R_n \sim \mathcal{N}(\mu, \sigma^2) \; with \; \mu = Y_n \; and \; \sigma = std_{Y_n} \; for \; n \in [1, N]$$

So, for each annual maximum, we can then randomly get a value which gives us one set of N annual maxima values. We then repeat this operation 10 000 times to get 10 000 sets of N annual maxima randomly obtained. We then fit a GEV distribution for each set which ultimately gives us 10 000 RLs curves randomly drawn. This is what we call the ensemble spread where we extract the 95th percentile to get a reasonable uncertainty spread.

This method is further compared with the commonly used GEV approach applied directly on the N-year annual maxima of the predicted mean values from the RF model which we simply referred as RF.

## 3 Results and discussion

### 3.1 Model validation

To validate the models, GOFs are calculated (partly presented in Table 2). For the time series of daily maxima, roughly similar statistics are found for all datasets whether using the QRF or LR. In general, we find slightly (but not significantly) better *r* and RMSE values associated with the LR and slightly better *perc95-bias* for the RF (not shown). For the annual maxima, the 95th, 97th, and 99th percentiles sets, marginally higher *r* and lower RMSE values are found for the LR in nearly all cases, with a maximum difference of 4 cm for the RMSE (except for 4 simulations out of the 36 where RMSE values vary up to 10 cm towards the extremes) and 0.10 for the *r* value. Overall, RMSE values are between 10 and 40 cm, and *r* values are between 0.4 and 0.9 in most cases when looking at the extreme sets. Smaller RMSE ranging from 5 to 15 cm and *r* values above 0.75 are found when directly looking at the predicted time series. Hence, error metrics are in general worse in both methods when calculated for extreme values (annual maxima and high percentiles) compared to overall values calculated from the full time series of predicted daily maxima values. For extremes, represented by the high percentile datasets, bias values range from -30 to -2 cm i.e., an underestimation of the observed extreme values for both the LR and RF. As shown in Table 2, biases vary with a maximum difference between models of ~10 cm in almost all cases. This highlights the fact that both models lose accuracy to predict ESLs compared to predicting less extremes events. And this seems to be caused by the non-linear effects occurring during the extremes as the decrease of *r* shows. Figure 3 depicts the correlation between observations and models in predicting Viken sea levels from the Hornbaek data trained on the 2010-2018 period. A similar picture is observed in nearly all cases (not shown). As shown in Fig. 3, the RF model returns significantly higher sea levels and shows higher variability towards the most extreme range than the LR, except when predicting Hornbaek sea levels from Viken data, where the model

is trained on observations from the 1990-2000 and 2000-2010 setup periods (not shown). In those two cases, the RF does not correctly reproduce the extreme range, as they are out-of-sample while the predicted values are bounded, since the RF can only reproduce in-sample events, but those correspond to the highest ESLs which the LR also struggles to reproduce and

205 therefore, the RF is not giving much lower values than the LR in this case. This effect disappears when the model is trained on a longer time period as we could see when investigating at the 20- and 30-years setup periods (not shown). Compared to an LR, it is clear that the inherently non-linear RF is better able to account for the few moderate extremes that occur during the 8-year training period, whereas they are likely to be suppressed in a linearized model.

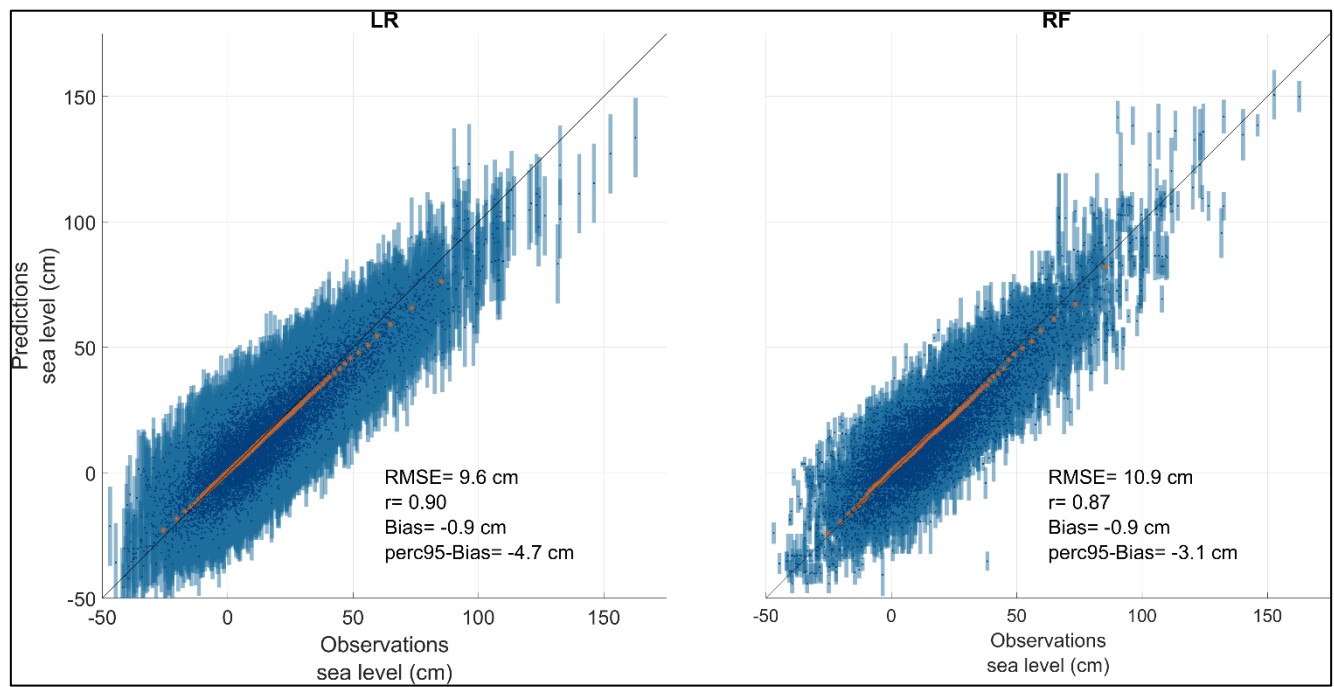

**Figure 3: Scatter plot between observations and LR (a) and RF (b) for predicting Viken sea level (y) from the Hornbaek tide gauge (x) for the setup period 2010-2020. Blue points show the daily maxima and each corresponding blue bar shows the standard deviation (b). Coloured stars correspond to the 1st, 2nd, 3rd , 4th, … to 99th percentile values of the dataset.**

Table 2 analyses the sensitivity to the distance between the two tide gauges for the 2010-2020 period. When the distance between two stations grows, the accuracy of both models seems to decrease, especially towards the extremes. For example, when looking at values on daily maxima but also on the extreme sets, RMSE values of around 8 to 20 cm and *r* values of around 0.7 to 0.9 are found for the sets reproducing Viken sea level from the Hornbaek tide gauge and its mirror set (9 km apart) compared to RMSE values of around 12 to 40 cm and *r* coefficients around 0.3 to 0.8 when reproducing Ringhals sea

levels from Hornbaek data and its mirror set (130 km apart) (Table 2). The highest differences are also observed for the extremes (annual maxima or 95th, 97th and 99th percentiles). Similar results are found when comparing the sea level time series

for Hornbaek and Ringhals, based on Viken data, and when comparing predictions for Viken and Hornbaek sea levels based on Ringhals data (not shown). Comparing results from mirrored sets, for example, when predicting Viken sea levels from Ringhals data or predicting Ringhals sea levels from Viken data, we do not always find the same performance, especially towards the extremes, as measured by the GOFs. This can probably be explained on physical grounds due to localised phenomena resulting for example from the topography or the local meteorological conditions, however, this is beyond the current technical note. Indeed, there are two sets of stations with very significant geographical differences. Hornbaek and Viken stations lie inside of a channel (almost at the entrance). However, compared to this stations, other stations are on the open coast. In general, the RF method seems to be more accurate than the LR when predicting ESLs, where it is essential to capture the non-linear behaviour and variability associated with the complex natural interactions between drivers of ESL events. These are likely to become more prominent when observations are obtained at sites further away. Conversely, an LR is inherently constrained by a linearity assumption.

**Table 2: RMSE and biases between different datasets evaluated in the validation period(s). Noticeable improvements ( >5 cm ) in terms of model bias with respect to annual maxima using the RF model or LR model are highlighted in bold or underlined respectively. A negative bias corresponds to an underestimation of the predicted values, and a positive one (italic) an overestimation. Error metrics calculated over the testing period for the case study of Halmstad city are displayed in bold italic. Because of the short length of the testing period, we do not calculate the bias on the annual maxima.**

| Predictor station x | Predictand station y | Setup period | *RMSE* on daily maxima (cm) | | *Bias* on daily maxima (cm) | | *Bias* on annual maxima (cm) | |
|---|---|---|---|---|---|---|---|---|
| | | | *LR* | *RF* | *LR* | *RF* | *LR* | *RF* |
| Viken | Hornbaek | 2010-2020 | 10.2 | 11.6 | *0.9* | *0.9* | -5.3 | -3.7 |
| Ringhals | Viken | 2010-2020 | 8.7 | 10.3 | -0.5 | -0.8 | **-28.4** | **-12.3** |
| Hornbaek | Ringhals | 2010-2020 | 12.1 | 14.3 | -1.0 | -1.1 | -17.9 | -21.1 |
| Viken | Ringhals | 2010-2020 | 8.7 | 10.3 | *0.5* | *0.4* | -6.5 | -8.7 |
| Ringhals | Hornbaek | 2010-2020 | 12.8 | 14.9 | *1.0* | *0.7* | **-30.6** | **-23.0** |
| | | 2010-2020 | 9.6 | 10.9 | -0.9 | -0.9 | **-17.5** | **-9.1** |
| | | 2000-2010 | 9.2 | 11.2 | -1.0 | -0.9 | **-18.4** | **-10.8** |
| | | 1990-2000 | 9.1 | 11.1 | *0.2* | -0.03 | -14.3 | -11.8 |
| | | 1990-2010 | 8.8 | 10.6 | *1.8* | *1.7* | **-12.0** | *6.6* |
| | | 2000-2020 | 9.8 | 11.2 | -1.1 | -0.9 | **-18.3** | **-7.1** |
| Hornbaek | Viken | 1990-2020 | 9.6 | 11.3 | -0.8 | -0.7 | **-21.9** | **-15.7** |
| *Hornbaek* | *Halmstad* | *2010-2020* | *8.9* | *11.8* | *1.0* | *1.1* | | |
| *Viken* | *Halmstad* | *2010-2020* | *6.2* | *7.7* | *0.8* | *0.8* | | |
| *Ringhals* | *Halmstad* | *2010-2020* | *7.5* | *8.8* | *0.7* | *0.9* | | |

## 3.2 Halmstad

The highest sea level recorded in Sweden occurred in Halmstad, indicating that Halmstad is highly susceptible to ESLs. However, the length of the local sea level time series is very short. Subsequently, the three stations: Hornbaek, Viken and Ringhals, are used for reconstructing the Halmstad sea level time series (Table 2). As shown above, using a RF or LR method, we can in principle reconstruct Halmstad sea levels back until 1891 for the period before observations became available in 2009 with reasonable confidence, using the station Hornbaek as a predictor, since this has the longest observed time series.

Because of the short length of Halmstad's time series, the training period is almost identical to the full time series which in practice makes it difficult to assess the model behaviour on extremes. Therefore, we used different 2-years of testing and 8-years of training periods to analyze how the model behaves for Halmstad station (setup period 2010-2020 with different testing periods: 2010-2012 / 2011-2013 / 2012-2014 / ... / 2017-2019 / 2018-2020) and this has been done for predicting Halmstad sea level from Hornbaek, Ringhals and Viken separately. Overall, the difference between each testing period is rather small with RMSE values differing from 1.5 to 4.1 cm; $r$ differs of around 0.03 to 0.06; bias of 5.2 to 6.8 cm; the perc95-bias differs more with 5.4 to 16.5 cm. We found out that, as in the other simulations done (predicting Viken from Hornbaek for example), the RF visually (from correlation plots not shown) behaves better towards the extremes (at least slightly) than the LR for all sets and tests except for the testing period 2015-2017 when predicting Halmstad from Viken or Hornbaek. Also, between LR and RF, RMSE values only vary from maximum 2.9 cm (xHornbaek / yHalmstad) and 1.9 cm in the 2 other sets. The xViken / yHalmstad set has the lowest RMSE values with an average across the simulations of 6.4 and 7.9 cm, bias of -0.2 and -0.4 and perc95-bias of -2.7 cm in both LR and RF respectively. Therefore, it seems the model behaves relatively well on extremes for Halmstad station even though we cannot fully assure its behaviour because of the short length of observations. Also, this conclusion is partly reinforced by the analysis between surrounding stations where the testing could be done over a larger time period using the 2018-2020 period as testing with the 2010-2020 as setup period.

In previous studies, Halmstad's RLs were calculated for current and future climate scenarios based on reconstructed sea levels from local wind speed observations of the Nidingen offshore station and Viken tide gauge data (Andersson, 2021). For Halmstad, RLs based on extended time series using the three neighbouring stations permit a reduction of the 95[th] percentile confidence interval (CI) compared with observations. Here, the full-period length of Halmstad's observed values (station y) are concatenated with the predicted time series to get the longest and more accurate extended time series possible before a GEV fit is applied. Even so, RLs are still lower even though within the uncertainty range than the ones displayed by Andersson (2021), which is a good sign, except for the 200-year RL with Viken as predictor when based on the RF mean outputs (fig. 4-a; Table 3). This can possibly be explained by the underestimation found towards the extremes on the predicted and therefore extended time series. This is why we introduced the RF method with random sampling which permits to represent more extremes values.

**Table 3: Halmstad's RLs from reconstructed time series using the outputs from the RF and the RF method with random sampling applied from the station Hornbaek (italic) compared with assessment by Andersson (2021) in bold italic.**

| Predictor: Station x | 5-year RL (m) | 10-year RL (m) | 50-year RL (m) | 100-year RL (m) | 200-year RL (m) | |
|---|---|---|---|---|---|---|
| **Hornbaek** | 1.6 | 1.8 | 2.1 | 2.2 | 2.3 | |
| **Ringhals** | 1.6 | 1.8 | 2.3 | 2.5 | 2.7 | RF |
| **Viken** | 1.6 | 1.7 | 2.0 | 2.1 | 2.2 | |
| **Viken + wind** | *1.8* | *2.0* | *2.4* | *2.6* | *2.8* | *Andersson* |
| **uncertainties** | *1.5 - 2.0* | *1.7 - 2.3* | *2.0 - 2.8* | *2.1 - 3.0* | *2.3 - 3.3* | *(2021)* |
| *Hornbaek* | 1.7 | 1.9 | 2.4 | 2.6 | 2.8 | *RF method with random sampling* |
| *95th percentile ensemble spread* | *1.6 - 1.8* | *1.8 - 2.0* | *2.2 - 2.7* | *2.3 - 3.0* | *2.5 - 3.3* | |

Conversely, we apply a RF-based random sampling to predict RLs probabilistically as described in section 2.2.4 (fig. 4-b), at Hornbaek, Viken and Ringhals (which permits extended time series of around 120 years, 35 years and 45 years, respectively). As would be expected due to the long time series, estimates based on Hornbaek data deliver the best performance and yield what seems like a reasonable 95th percentile ensemble spread (Fig. 4). The inferred RLs are slightly higher than the RLs derived directly from observations, which are associated with a very large 95th percentile CI due to the short length of the time series. The predictions using Viken data present the lowest RLs with a 95th percentile ensemble spread (upper values) corresponding almost to the median RLs from observations probably underestimating the extremes. On the other hand, predictions from Ringhals result in the highest RLs but like Viken are also associated with a rather large ensemble spread. Because of the lengths of the respective time series, there is low confidence in return periods of rare occurrences such as a 200-year event (a little less pronounced for Hornbaek-based predictions). This challenge of rare occurrences is evident when looking at the 95th percentile CIs for each RL curve resulting from the RF method with random sampling. For Halmstad, RLs based on inputs from the Hornbaek station following the RF method with random sampling are close to the ones displayed by Andersson (2021), highlighting the importance of considering the full uncertainty range when predicting high sea levels from a small sample of such events (Table 3).

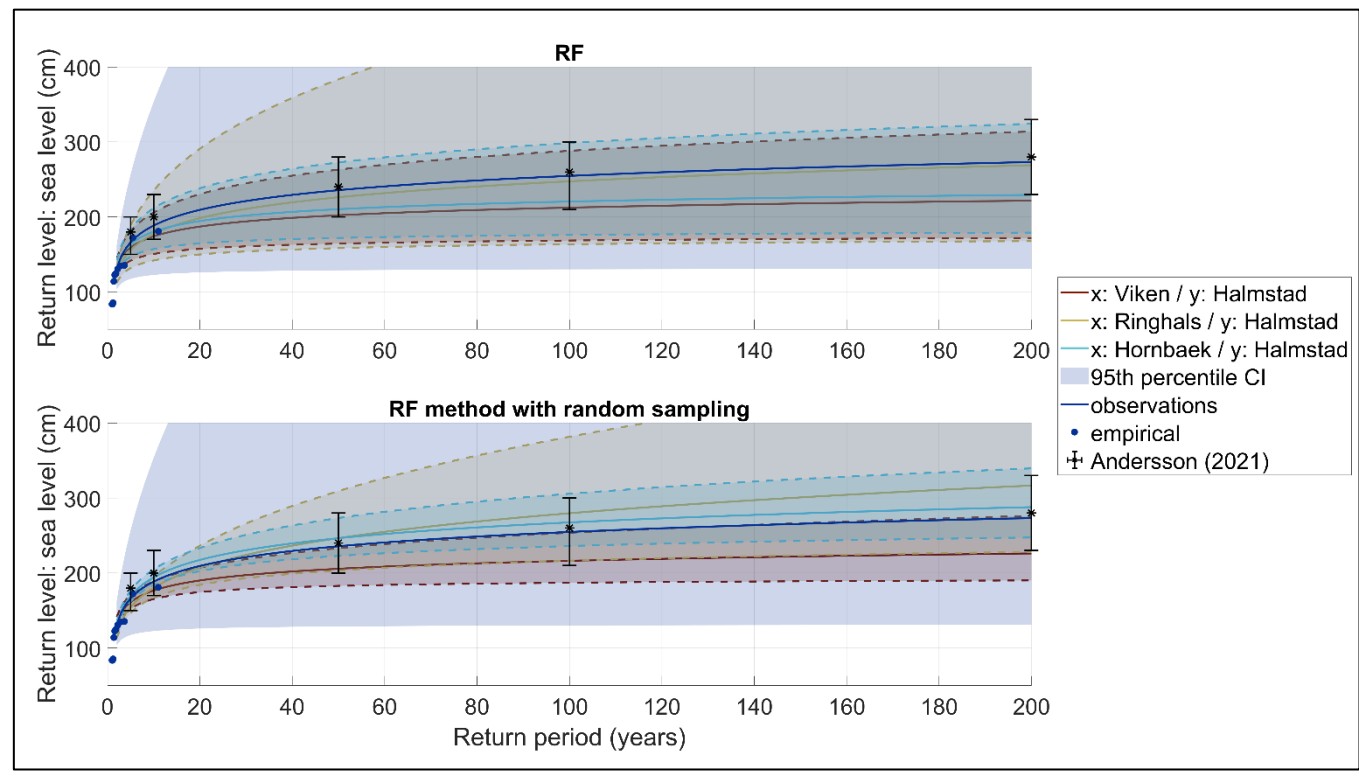

**Figure 4: RLs from each reconstructed time series to predict sea level for Halmstad based on RF model mean outputs (a) and following the RF method with random sampling (b). It displays Maximum Likelihood Estimates of the GEV distribution fits of each dataset associated with 95th percentile confidence intervals (CIs) (a) and 95th percentile ensemble spread (b) in background colours, and the dots are the empirical data from observations. Black error bars show RLs and 95th percentile CI calculated from Andersson (2021).**

## 4 Limitations

It is evident that our statistical reconstructions are limited by several factors in particular local ocean dynamics and the length of the time series used. Both are especially important for extreme analysis. We implicitly assume that a time window of only 10 years is sufficient for describing the relationship between two stations under normal ocean conditions, and while this study seems to support this hypothesis, it is by no means assured that this will be the case for any two neighbouring stations. Especially when the relationship is found to be highly non-linear. For non-normal situations like ESLs, it is evident that our setup period is principally much too short to learn the (inherently non-linear) dynamics related to rare sea level extremes, and that our modelling essentially yields an extrapolation of the normal ocean dynamics relating two sites, which may introduce significant biases in the subsequent RL estimates. This limitation is general for most if not all types of extensions of observed time series using neighboring data. Even, so it is trivial to assume that non-linear and non-parametric methods like the RF outperform other methods in terms of capturing extreme trends within a very short time window.

As indicated earlier, the RF is limited in range by the input values. Hence, in principle, this method is not suitable for extrapolating to higher values than what is seen in the training period, as highlighted when predicting Hornbaek sea levels from the Viken tide gauge based on the 1990-2000 and 2000-2010 setup periods. This limitation is a known issue when applying random forests-based prediction models (Tyralis et al., 2019; Hengl et al., 2018); it can be mitigated to some extent by using many extended time series for model training as new data becomes available. In this study, we did not find out-of-sample issues to have a strong influence as the RF model reproduced extremes rather well. Adding additional sources, e.g. observed wind information, may also improve predictions (Johansson et al., 2018) or reanalysis (Hieronymus et al., 2019). However, these approaches were outside the scope of this technical note, focusing on exploring the limitations and advantages of only using neighbouring observations of sea level. If more complex methods can achieve additional accuracy, this is of course of great value, but it may also confuse the interpretations at times, which is not preferable. In preliminary tests, additional improvements from adding reanalysis and hindcast data did not appear to add enough value to warrant the decreased interpretability, but this is certainly a promising research area.

Finally, this study focused on a limited area of the Swedish West Coast. The methodology is generally applicable but contingent on local conditions, and hence further research is needed to investigate if similar performance can be found when applying the proposed method to other areas with different ocean dynamics.

## 5 Summary and conclusions

This study demonstrates that sea level time series of daily maxima can be relatively successfully reconstructed from a neighbouring station using the LR or RF approaches using even very short overlapping intervals (10 years). As expected due to the short length of the overlap, ESLs are somewhat underestimated. The RF model is better able to capture the inherent non-linearities and hence proves to be more accurate during those conditions. The corresponding absolute bias values are generally lower than those found from the LR. The best reconstructions are generally achieved for stations spatially closer to each other, though this can be partially offset using the RF, which is found to yield better results than the LR for stations spatially located further away from each other. We tested another method that we named "RF method with random sampling" in the case of Halmstad. When applied to reconstructed time series from a 10-years dataset, the method confirmed the results from a previous more physics-based study, reproducing RLs with a reasonable uncertainty range given by the 95[th] percentile ensemble spread. The method is easily applicable to other sites and can also be applied across regions as long as two neighbouring stations' sea-level time series are available. Overall, using the RF method with random sampling to represent the uncertainty of extremes could be an advantage compared to many single-output machine learning predictions.

*Code availability.* The code used to generate the figures and tables can be acquired by contacting the first author (kevin.dubois@geo.uu.se).

*Data availability.* The Hornbaek data used are available upon request. The data from the different Swedish stations are
340 available at https://www.smhi.se/data/oceanografi/ladda-ner-oceanografiska-observationer#param=sealevelrh2000,stations=core (last access: 14 October 2021).

*Author contributions.* KD developed the code and conducted the analysis. KD prepared the manuscript with contributions from all co-authors.

*Competing interests.* The authors declare that they have no conflict of interests.

*Acknowledgements.* The work forms part of the project: Extreme events in the coastal zone – a multidisciplinary approach for better preparedness.

*Financial support.* This research was funded by the Swedish Research Council FORMAS (Grant No. 2018-01784) and the Centre of Natural Hazards and Disaster Science (CNDS). AR and EN were also partially funded by the Research Council of Norway, MachineOcean project 303411.

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
