# Peer review of "Technical note: Extending sea level time series for extremes analysis with statistical methods and neighbouring station data"

_EGUsphere, 2023_

## Referee Comment (RC2)

**Comments on "Technical note: Extending sea level time series for extremes analysis with machine learning and neighbouring station data"**

Anonymous referee #2

September 8, 2023

Note: quotes from the pre-print are in blue.

**1  General comments**

The preprint discusses the artificial extension of a short (10 years) tide gauge record in the city of Halmstad, using longer records from neighbouring stations, in order to better estimate return levels of extreme sea levels in Halmstad. To do so, a statistical relationship between the return levels of Halmstad and its neighbouring stations is found in the overlapping 10-year period through linear regression and quantile regression forest.

I think the paper addresses a complicated, yet very interesting issue. The extension of a short record to better estimate extreme return levels is a delicate task, both from the statistical and physical/operational point of view. I think the paper lacks discussion on the limitations of the study, although many important issues are already mentioned in the text. Also, I believe that the explanation of the statistical method is somewhat confusing, and could be made much clearer for the reader. Suggestions in this direction are given in the specific comments.

For these reasons, my opinion leans toward a major review of the paper before it can be accepted.

**2  Specific comments**

(a) Lines 32-33: This technical note evaluates a machine learning (ML) method for extending the sea level time series obtained by a tide gauge of interest using a longer time series at a neighbouring tide gauge in the context of analysing sea level extremes.

This is particularly challenging from a statistical point of view. You should discuss more the hypothesis behind this work. You should always keep in mind that your objective is to have a more robust estimation of ESL based on your extrapolation method.

Which ESL can your algorithm reproduce ? Only the following ones:
- ESL that were observed during the short period (since LR is mostly driven by "average" values, and QRF cannot reproduce out-of-sample values)
- ESL that are associated with a trace on other tide gauges

Therefore, your algorithm cannot reproduce:
- ESL that were never observed (and it is very likely that they happened during the period that you try to simulate)
- ESL that leave no trace on neighbouring tide gauges.

The problem of "unobserved extremes" can be tested, as these might have left a footprint on the neighbouring tide gauges that is stronger than what was observed during the 2010-2020. If there are observed ESL of neighbouring stations outside the 2010-2020 period that are stronger than the ones observed in the 2010-2020 period, then it is likely that Halmstad also encountered extremes stronger than the ones of 2010-2020.

The issue of ESL leaving no trace on the neighbouring stations could also be tested, or at least discussed: did you see any ESL in the 2010-2020 period that the QRF and/or LR failed to reproduce ? What would they be linked to ? I suppose it would be very local events ?

These points are crucial and were not tested, I think you should test them or at least mention them clearly.

On the same topic: your confidence intervals on RLs will decrease as you use the extended time-window with sea-levels predicted from neighbouring stations (e.g., Figure 4). However, is this reduction justified, based on the previous observations ? It seems to me that the added error inherent to LR and QRF is not clearly included in your evaluation of RLs: it seems like the data you add is given the same value as the real observations. Please correct me if I am wrong on this point.

(b) *(This comment is related to the previous one.)* End of Table 2's caption:
Because of the short length of the testing period, we do not calculate the bias on the annual maxima.
Also lines 176-178:
As shown above, using a QRF or LR method, we can in principle reconstruct Halmstad sea levels back until 1891 for the period before observations became available in 2009 with reasonable confidence, using the station Hornbaek as a predictor, since this has the longest observed time series.

Since you cannot calculate the bias on annual maxima, what are your guarantees that your method will be able to reproduce extremes in the past ? I suppose it would be available from the bias on daily maxima ? However, in Table 2, the line with the largest computed biases on annual maxima is Ringhals-Hornbaek, and this line shows pretty low values of biases on daily maxima, suggesting a weak link between the biases on daily maxima and on annual maxima. Similarly, high values of $r$ are not incompatible with high values of annual maxima, as confirmed by the Ringhals-Viken example. However, since we do not know if a few tens of centimeter is a large value, we are not able to assess, as a reader, if the method can be trusted.
You should explain why you have confidence in extrapolating towards extremes in the past. This demands statistical tests that you could try to do on your short sample (12 years ?). For instance, you could train/validate your model on 10 years and then test on the 2 remaining years, and repeat this operation by switching which years are used for training/validation and which years are used for testing. This is classical in machine learning. This would allow to evaluate, at least a little bit, how your model behaves on extremes for the Halmstad station. Lines 186-188: "When comparing the QRF method to LR, slightly better RMSE and r values are found for the LR, but when looking at higher percentile levels, the QRF results in higher corresponding values than for LR in all sets (not shown)."
This piece of information is probably more interesting than the $r$ values showed in Table 2. I think you should focus on that.

(c) I believe that the expression "machine learning" is used in a confusing way throughout the manuscript. The QRF method is called "machine learning" in opposition to the LR, but in fact the LR is also a machine learning technique, only perhaps simpler and much more common than QRF. I therefore suggest that the expression "machine learning" be used, but only in the introduction and conclusion, and it should refer to *both* methods, LR and QRF. I recommend that the acronym "ML" be withdrawn from the manuscript and replaced by "QRF" to avoid ambiguity.

(d) Lines 85-87: Would the results change if the validation period was different (i.e., two years in the beginning instead of the end, or even two random years picked in the 10 years). Testing the sensibility to this choice would make the results even more robust.

(e) Lines 91-92: Based on each x/y predictor-reconstruction station pair, a linear equation is found using the least squares method as means of determining the best fit coefficients.

It could be nice to see the coefficients and try to interpret their meaning : I assume they would be positive (high sea level at one station means high sea-level at another station) and would reflect both
- the ratio of intensity of sea-level variations between stations
- the correlation between stations.

Even if you choose not to show the values of the coefficients in the final version of the article, I would like to see them as a sanity check, and I believe a little comment on our interpretation of the coefficients would be nice in the manuscript, even if you do not show them.

(f) Lines 92-93: It feels like the predictor for the linear regression and QRF could have been chosen to be a little bit more complex, revealing other time-space relationships between the stations. Since the sea level is sensitive to meteorological conditions, which are advected by the winds, the sea level at one station at time $t$ might be better predicted if using time-lagged sea-level from another station, e.g. the sea-level at times $t\pm$ a few days (this could be tested very easily from your algorithm, at least for a couple of stations). Even time-delayed embeddings could be used, with sea-level at times $t - m$days, ..., $t - 1$day, $t$, $t + 1$day, ..., $t + n$days where $m$ and $n$ would have to be optimized. Although this might be out of the scope of the study, it should be tested or at least mentioned. But maybe you have reasons to believe that this would not be useful/necessary ?

(g) I believe section 2.2.2 and section 2.2.4 need to be clarified. Although I am not an expert of QRF, it seems to me that your formulation is misleading.

Lines 96-97 The QRF method yields a mean and a standard deviation for each predicted value (Breiman, 2001; Meinshausen, 2006. You should add equation to this sentence to make it clear. QRF estimates quantiles of a predictand, not average and standard deviation, therefore your formulation is confusing. What do you call "predicted value" ? You need to be specific and to use precise vocabulary. What you should do is to describe the method in detail, to help unfamiliar readers. I suggest to extend this paragraph and to add equations.

Lines 97-98 The QRF model is implemented using the MATLAB function TreeBagger where the regression method is based on a number of trees and minimum leaf size hyper-parameters. Reading the MATLAB documentation indicates that TreeBagger cannot be used alone, another function must be used to perform the regression. Did you use "quantilePredict" ? "predict" ? "fitrensemble" ? Something else ? Please specify this to allow reproducibility and help understanding.

Line 98 These parameters are here set to 500 and 1, respectively. Consulting the documentation on TreeBagger MATLAB method, it seems that 1 is the default parameter for classification trees. Some justification for the choice of 500 would be nice here.

Line 219 QRF method with random sampling to evaluate return levels (RLs) "QRF method with random sampling" is not a known terminology, it is one you designed for the purpose of this study. Therefore, it must be made very clear in the manuscript. For instance, you could say "in the following, we denote "QRF method with random sampling" the following methodology:..." and then describe your methodology. The description must be very clear and thorough, including every step of the calculation, to avoid misunderstanding and allow reproductibility of your results.

Also, since this method is compared with another way of estimating return levels (Figure 4) you should explain this other method of estimating RLs (simply named "QRF" in Figure 4) in this section as well.

Lines 120-122 Based on the QRF daily means and standard deviations, we calculate the corresponding annual maxima from the reproduced time series and their associated standard deviations. This isn't clear to me. I suppose "the corresponding annual maxima" are the annual maxima of average QRF predictions ? But how do you incorporate the standard deviations in the maxima ? You should write equations for this.

Lines 122-123 We assume that a Gaussian distribution describes the probability for each predicted annual maximum. It seems that this is a hypothesis that you could (and should) test for.

Line 123 10 000 sets of 30-year maxima You have Gaussian distributions for annual maxima, and you use it to draw 30-year maxima ? Why 30-year maxima ? What does this mean ?

Line 125 This yields an ensemble of randomly drawn RL curves. Why would you trust this method rather than simply using the QRF-mean daily maxima ? If one method is better from a statistical viewpoint, then there is no point in doing both (and showing both in the report). This adds confusion. I suggest you consider keeping only one, either "QRF" or "QRF random sampling". If not, this choice should be motivated.

Due to this confusions, Figure 4 appears unclear to me, while it is the most important figure of the pre-print. Maybe it is due to my lack of knowledge of QRF methods, but I doubt that this is the only reason. Anyway, this technical note should be accessible for readers unfamiliar with QRF.

(h) Figure 3 Since you show only one example, I think it would be better to show one with Halmstad as predictand, as this is the main objective of your study. This would also allow you to illustrate the points mentioned here in comment (a).

**3   Technical corrections**

1. I think using only "ESL" and not "ESLs" would be enough, and clearer. However this decision is yours to make.

2. Figure 1: Some of the fonts are too small to be read (the latitudes/longitudes, as well as the city names on the left panel). Either enlarge the font or supress the text.

3. Line 71: from which the annual (daily) maximum $\rightarrow$ you could remove "(daily)".

4. Lines 76-77: Conversely, long-term linear trends (i.e., sea level rise) were estimated for all time series and found to range between 0.34 and 1.47 cm per decade..
   Could you indicate all values of computed linear trends, along with the corresponding city ? Since there are only 4 stations it would not be excessively lengthy. Also, I think the way these linear trends are estimated should be explained in a bit more details. There are different ways of estimating linear trends for sea-level, corresponding to different hypothesis. In particular, for the Hornbaek station, is the linear trend computed using the whole time series (i.e. before 1900) ? Is this relevant or should the rise start later ? Does it make any difference for the estimated time-series ? Although it might not make a huge difference, it seems important to indicate this, since you are estimating long return periods and since Hornbaek is the longest time-series in your dataset, and therefore the time-series which contains a large part of the information on which your ML techniques rely.
   "It is worth noting that since we use observed tide gauge data, long-term trends, that is, climate change induced sea level rise are implicitly considered, although site-specific changes in the relative sea levels due to, e.g., human activities may introduce biases."
   Same here. These points are crucial to your study and need to be debated more.

5. Lines 83-84: The proposed approach for extending short sea level time series uses one neighbouring station as predictor (station x) of past sea level data at the station of interest (station y). The way x and y are defined is not fully clear. I assume that you are using daily maxima, with long-term linear trend removed. Please recall this here.

6. Lines 92-93: the sea level at station $y$ is predicted from the sea level at station $x$. Perhaps you should make it clearer that it is the sea level at station $y$, time $t$, that is predicted from the sea level at station $x$, time $t$. See comment above for the same lines 92-93.

7. Table 3: You provide "uncertainties" from the paper by Andersson (2001), however, I cannot find an explanation of what these uncertainties are, more precisely. This would help to compare it with your "95th percentile ensemble spread".

8. Also in Table 3: I think you should be able to give uncertainties associated with the QRF-based RLs from every station, and therefore add a line of the type "uncertainties" below each line. This is a type of output available from a QRF model I believe.

9. Also in Table 3: Are the RLs computed by Anderson based only on Viken as predictor, or are the winds also used as mentioned in the text ? If this is the case, you should specify it in the table with something like "Viken + wind". If not, you should specify it in the text to avoid confusion. Also, I think a bit more description of Andersson's method would be helpful here, since you use it many times for comparison.

10. Lines 193-194:
"Here, observed values are added to the extended time series to get the longest time series possible before a GEV fit is applied."
You should be more specific. Which observations are added ? How many years/months ? How much does that strengthen your model ?

11. Lines 194-195:
"Even so, RLs are still lower than the ones displayed by Andersson (2001) when based on the ML mean outputs (fig. 4-a; Table 3)." How could you explain this systematic bias ? What does it reflect ? Also, I think you should mention that the estimated RLs in Table 3 are all in the uncertainty range of Andersson's study (2001), which is a good sign, except for the 200-year RL with Viken as predictor.

12. Line 130 RMSE RMSE is not shown → perhaps add it to the table, in [cm]. It would give an idea of the relative importance of the biases, which is not clear here: a few centimeters seems to be small, but if we don't know the amplitude of typical variations of sea-level there is no way to really know (and this information is station-dependant). OR you could show the relative biases in the Table (for instance: baises normalized by RMS sea-level decadal variations around the mean)

13. Line 131 *perc95-bias* I do not see this in Table 2 ?

14. Lines 131-133 For the annual maxima, the 95 th , 97 th , and 99 th percentiles sets, marginally higher r and lower RMSE values are found for the LR in nearly all cases, with a maximum difference of 6 cm for the RMSE and 0.10 for the r value Not shown. Also, how can that be understood together with the fact that the biases are somewhat smaller when using the QRF ? It seems counterintuitive, this should be explained.

15. Lines 133-134 Overall, RMSE values are between 10 and 40 cm, and r values are between 0.6 and 0.9 in most cases. what does that indicate ?

16. Line 135 a slight underestimation of the observed extreme values for both the LR and QRF how do we know that -30cm is "slight" ?

17. Line 138 is observed in nearly all cases add "not shown".

18. Lines 140-144 In those two cases, the QRF does not correctly reproduce the extreme range, as they are out-of-sample while the predicted values are bounded, since the ML can only reproduce in-sample events. Compared to an LR, it is clear that the inherently non-linear QRF is better able to account for the few moderate extremes that occur during the 8-year training period, whereas they are likely to be suppressed in a linearized model. To me, this is a very interesting point here. You seem to conclude that the QRF is better than the LR, since the latter smooths out the extremes, however you also point out that the QRF is not able to produce out-of-sample predictions. I would recommend a more nuanced conclusion.

19. Figure 3

    - Standard deviation is also available when performing LR (it is assumed to be always the same, this is called homoscedasticity), it is given by the RMSE, therefore you should also plot the error bars for the LR.

    - It seems like you are not using QRF but simply RF, since you estimate a mean and standard deviation, am I mistaken ?

    - I see coloured stars close to the diagonal y=x but I do not see how these could be 1st and 99th percentiles ? This part isn't clear.

    - The figure is quite fuzzy in the dense area of average sea levels between 0 and 50cm. I recommend that you do not show all error bars, perhaps only for extremes (i.e., above/below certain quantiles) as this is the main objective of your study.

20. Line 152 the model accuracy clearly decreases It does not seem "clear" to me. For instance, the best r values (0.91, LR) are found for the pair Viken-Ringhals and Ringhlas-Viken, which seem to be pretty far apart on the map. Only a more systematic study of the statistics (r, biases, etc.) with respect to the distance between the cities would reveal undoubtedly this distance-dependency (which is probably true).

21. Line 153 around 0.7 I don't see this value in Table 2 ?

22. Line 153 (9 km apart) perhaps show distances somewhere in the Table to improve readability ?

23. Line 153 r coefficients around 0.3 I don't see the value "0.3" in the table ?

24. Lines 155-156 (annual maxima or 95 th, 97 th and 99 th percentiles). add the mention "not shown"

25. Lines 156-157 imilar results are found when comparing the sea level time series for Hornbaek and Ringhals, based on Viken data, and when comparing predictions for Viken and Hornbaek sea levels based on Ringhals data. add "not shown"

26. Lines 159-160 This can probably be explained on physical grounds however, this is beyond the current technical note. There is nothing we can do with this information, I think it should not be specified, or you should say more.

27. Lines 160-161 In general, the QRF method seems to be more accurate than the LR when predicting local sea levels from stations located further away e.g., between Ringhals and Viken / Hornbaek as compared to Viken and Hornbaek Should we see this in the annual maxima ? It is not true when we look at the predictions of Ringhals based on Viken, or Hornbaek based on Viken (200-2010, 5cm stronger bias for the annual maxima using QRF). **More generally, you should always mention which numbers in the Table support your claim otherwise it can be questioned.**

28. Table 2 About the colors, is it the right choice ? You do not indicate why the sign is so important that you highlight it in colour. Red highlighting connotes "danger", should we fear an overestimation of sea-level ?

29. Table 2 You highlight in bold when QRF is better than LR in bias on annual maxima by 5cm. To be fair, you should also highlight (differently) when LR is better than QRF by 5cm, this is the case for the couple Viken-Hornbaek 2000-2010.

30. Line 183 the setup period replace by "the train and validation period"

31. Line 184-185 When analysing the error metrics over the testing period, the model based on Viken station presents the best results. What makes you say that ? The biases are smaller with the Ringhals station as predictor.

32. Line 189 (not shown However this is probably the most important piece of information !

33. Lines 193-194 Here, observed values are added to the extended time series to get the longest time series possible before a GEV fit is applied. What are these added values ? You have to be more specific for readers to know what you have done.

34. Table 3

    - Station x you could add "predictor".

    - 5th line, for the Andersson (2021) study, you write simply "Viken", but I understand that winds are also used to make this estimate ? Therefore you should perhaps write "Viken+winds" in the first column, 5th line of the Table.

35. Line 203-204 The inferred RLs are slightly higher than the RLs derived directly from observations are these observation-based RLs shown anywhere in the paper ?

36. Figure 4's caption (end) Black error bars show RLs and 95 th percentile CI calculated from Andersson (2021). → this should also be in the legend on the right.

    - also, what is "MLE" in the legend ?

    - also, all elements of the legend should be in the same box, here it seems like the upper box is for the upper panel and the lower box for the lower panel, which is not the case

37. Lines 232-233 This limitation is a known issue when applying ML-based prediction models (Tyralis et al., 2019; Hengl et al., 2018); Wrong use of "ML" : many machine learning algorithm can produce values outside of the observed range. The two cited paper are about random forests, not ML in general.

38. Line 239 but it may also confuse the interpretations at times, could you be more specific ?

39. Line 241 but this is certainly an active research area I think you should replace "active" by "promising"

40. Line 250 The best reconstructions are generally achieved for stations spatially closer maybe this would change if you allow to use time-delays in the definition of $x$. See comment (f) above.

41. Line 252 We tested the QRF method with random sampling Replace by "We tested another method that we named 'QRF with random sapling'."

42. Line 281 That doi seems to point to another article which is not the one you mention in the text.

---

## Author Comment (AC1)

**Comments on "Technical note: Extending sea level time series for extremes analysis with machine learning and neighbouring station data"**

Anonymous referee #2

September 8, 2023

Note: quotes from the pre-print are in blue.

**1 General comments**

The preprint discusses the artificial extension of a short (10 years) tide gauge record in the city of Halmstad, using longer records from neighbouring stations, in order to better estimate return levels of extreme sea levels in Halmstad. To do so, a statistical relationship between the return levels of Halmstad and its neighbouring stations are found in the overlapping 10-year period through linear regression and quantile regression forest.

I think the paper addresses a complicated, yet very interesting issue. The extension of a short record to better estimate extreme return levels is a delicate task, both from the statistical and physical/operational point of view. I think the paper lacks discussion on the limitations of the study, although many important issues are already mentioned in the text. Also, I believe that the explanation of the statistical method is somewhat confusing, and could be made much clearer for the reader. Suggestions in this direction are given in the specific comments.

For these reasons, my opinion leans toward a major review of the paper before it can be accepted.

Thank you for your review and many helpful comments to improve our study. We agree with you that some additional clarification are needed on the methods we used and also on the limitations of the studied approaches and more generally to methods attempting to extend time series using longer records of neighbouring data. We hope our responses given below marked in red as well as our additional analysis and changes in the manuscript have helped to address these issues.

**2 Specific comments**

(a) Lines 32-33: This technical note evaluates a machine learning (ML) method for extending the sea level time series obtained by a tide gauge of interest using a longer time series at a neighbouring tide gauge in the context of analysing sea level extremes.

This is particularly challenging from a statistical point of view. You should discuss more the hypothesis behind this work. You should always keep in mind that your objective is to have a more robust estimation of ESL based on your extrapolation method.

Which ESL can your algorithm reproduce ? Only the following ones:

- ESL that were observed during the short period (since LR is mostly driven by "average" values, and QRF cannot reproduce out-of-sample values)

- ESL that are associated with a trace on other tide gauges

Therefore, your algorithm cannot reproduce:

- ESL that were never observed (and it is very likely that they happened during the period that you try to simulate)

- ESL that leave no trace on neighbouring tide gauges.

The problem of "unobserved extremes" can be tested, as these might have left a footprint on the neighbouring tide gauges that is stronger than what was observed during the 2010-2020. If there are observed ESL of neighbouring stations outside the 2010-2020 period that are stronger than the ones observed in the 2010-2020 period, then it is likely that Halmstad also encountered extremes stronger than the ones of 2010-2020.

You are right and we expanded the analysis a bit as described below and to some degree in the manuscript (see next comment). To check the problem of "unobserved extremes", we also decided to reconduct the analysis for all x/y station pair and for each setup periods with different time length: (10 years) 1990-2000 / 2000-2010 / 2010-2020 / (20 years) 1990-2010 / 2000-2020 / (30 years) 1990-2020 (table r3). However, to keep the paper as a technical note, we did not want to expand the manuscript too much and would rather like to keep it concise and focused to be more reader-friendly while still maintaining the most important messages explained well-enough. At the same time we fully agree with your point and have therefore added comments to convey this very important limitation (already discussed in the limitations section) as it is general for most if not all types of extensions of observed time-series using neighbouring data.

The issue of ESL leaving no trace on the neighbouring stations could also be tested, or at least discussed: did you see any ESL in the 2010-2020 period that the QRF and/or LR failed to reproduce ? What would they be linked to ? I suppose it would be very local events ?

Our analysis on the different stations show an underestimation in magnitude on the extremes which is mentioned in the paper with, most of the time, a bigger underestimation using the LR. However, in my opinion, no clear ESL in the 2010-2020 period failed to reproduce except for one case (x: Hornbaek / y: Ringhals) where an observed value of around 160 cm failed to get reproduced in both models with a predicted value of around 70cm for the setup periods (2010-2020 / 1990-2000 / 1990-2010). However, there is overall an underestimation of some events, but we would not call it a failure to reproduce this ESL. But, if this would be the case, we would agree with you and they would most likely be the result effects of really local conditions as some seiche effects or compound events conditions for example.

These points are crucial and were not tested, I think you should test them or at least mention them clearly.

We now mention more about limitations in the manuscript also please see other comments about the additional tests conducted.

On the same topic: your confidence intervals on RLs will decrease as you use the extended timewindow with sea-levels predicted from neighbouring stations (e.g., Figure 4). However, is this reduction justified, based on the previous observations ? It seems to me that the added error inherent to LR and QRF is not clearly included in your evaluation of RLs: it seems like the data you add is given the same value as the real observations. Please correct me if I am wrong on

this point.

This is indeed a challenging task and a good question. As we use extended data time-series, if we trust it, the confidence intervals on RLs decrease. However, as you mentioned, there is an added error inherent to LR and RF which is difficult to account for and this is why we decided to create the method we called "RF with random sampling". To clarify this method which is just proposed here but not validated, the idea is to use the information

from the RF based on a mean and standard deviation (std) value. Indeed, each predicted value is associated with a std which permits us to randomly generate values out of that range in assuming a Gaussian fit for each predicted value and std. Then, with a large enough sample size, each ESL can be associated with a statistically rare event, also for higher sea levels than the highest in a short record. We agree with you that the data added is however highly influenced by the observations in the short record and therefore influenced by the observed extremes on the site. This is reasonable and likely true for all statistical methods that attempts to extend short observational records. We have therefore as discussed later elaborated the analysis and sensitivity study for different time periods, as you suggested. We also come back to the later specific question about the estimated standard deviation and mean value in the RF predictions.

(b) (This comment is related to the previous one.) End of Table 2's caption:

Because of the short length of the testing period, we do not calculate the bias on the annual maxima.

Also lines 176-178:

As shown above, using a QRF or LR method, we can in principle reconstruct Halmstad sea levels
back until 1891 for the period before observations became available in 2009 with reasonable confidence, using the
station Hornbaek as a predictor, since this has the longest observed time series.

Since you cannot calculate the bias on annual maxima, what are your guarantees that your method will be able to reproduce extremes in the past ? I suppose it would be available from the bias on daily maxima ? However, in Table 2, the line with the largest computed biases on annual maxima is Ringhals-Hornbaek, and this line shows pretty low values of biases on daily maxima, suggesting a weak link between the biases on daily maxima and on annual maxima. Similarly, high values of r are not incompatible with high values of annual maxima, as confirmed by the Ringhals-Viken example. However, since we do not know if a few tens of centimeter is a large value, we are not able to assess, as a reader, if the method can be trusted. You should explain why you have confidence in extrapolating towards extremes in the past.

This demands statistical tests that you could try to do on your short sample (12 years ?). For instance, you could train/validate your model on 10 years and then test on the 2 remaining years, and repeat this operation by switching which years are used for training/validation and which years are used for testing. This is classical in machine learning. This would allow to evaluate, at least a little bit, how your model behaves on extremes for the Halmstad station. Lines 186-188: "When comparing the QRF method to LR, slightly better RMSE and r values are found for the LR, but when looking at higher percentile levels, the QRF results in higher corresponding values than for LR in all sets (not shown)." This piece of information is probably more interesting than the r values showed in Table 2. I think you should focus on that.

That is a fair point and, as you proposed, more testing has been done using different 2-years of testing and 8-years of training periods to analyze how the model behaves for Halmstad station (setup period 2010-2020 with different testing periods: 2010-2012 / 2011-2013 / 2012-2014 / … / 2017-2019 / 2018-2020) and this has been done for predicting Halmstad sea level from Hornbaek, Ringhals and Viken separately. The results are presented on the table r1 below with the minimum and maximum values resulted from the nine simulations (with each different testing periods) for each set predicting Halmstad. Overall, the difference between each testing period is rather small with RMSE values differing from 1.5 to 4.1 cm; r differs of around 0.03 to 0.06; bias of 5.2 to 6.8 cm; the perc95-

bias differs more with 5.4 to 16.5 cm. We found out that, as in the other simulations done (predicting Viken from Hornbaek for example, cf. Figure 3 in the manuscript), the RF behaves better towards the extremes (at least slightly) than the LR for all sets and tests except for the testing period 2015-2017 when predicting Halmstad from Viken or Hornbaek. Also,between LR and RF, RMSE values only vary from maximum 2.9 cm (xHornbaek / yHalmstad) and 1.9 cm in the 2 other sets. The xViken / yHalmstad set has the lowest RMSE values with an average across the simulations of 6.4 and 7.9 cm, bias of -0.2 and -0.4 and perc95-bias of -2.7 cm in both LR and RF respectively. Therefore, it seems the model behaves relatively well on extremes for Halmstad station. Also, this conclusion is partly reinforced by the analysis between surrounding stations where the testing could be done over a larger time period using the 2018-2020 period as testing with the 2010-2020 as setup period.

This has been added in section 3.2.

*Table r1. GOFs ranges for the different tests predicting Halmstad's sea level from Hornbaek, Viken and Ringhals.*

| sets | LR | | | | RF | | | |
|---|---|---|---|---|---|---|---|---|
| | RMSE (cm) | r | bias (cm) | perc95-bias (cm) | RMSE (cm) | r | bias (cm) | perc95-bias (cm) |
| xHornbaek/yHalmstad | [8.3 ; 12.4] | [0.88 ; 0.94] | [-2.2 ; 3.5] | [-11.9 ; 4.6] | [11.1 ; 14.6] | [0.83 ; 0.89] | [-2.7 ; 3.7] | [-9.7 ; 5] |
| xViken/yHalmstad | [6.1 ; 8.3] | [0.93 ; 0.98] | [-2.3 ; 2.9] | [-5.2 ; 2.5] | [7.5 ; 9] | [0.92 ; 0.96] | [-2.4 ; 2.8] | [-4.4 ; 1] |
| xRinghals/yHalmstad | [7.2 ; 8.7] | [0.93 ; 0.96] | [-3.9 ; 2.9] | [-8.7 ; 1.8] | [8.8 ; 10.3] | [0.91 ; 0.94] | [-4.2 ; 2.5] | [-10.9 ; 4.8] |

(c) I believe that the expression "machine learning" is used in a confusing way throughout the manuscript. The QRF method is called "machine learning" in opposition to the LR, but in fact the LR is also a machine learning technique, only perhaps simpler and much more common than QRF. I therefore suggest that the expression "machine learning" be used, but only in the introduction and conclusion, and it should refer to both methods, LR and QRF. I recommend that the acronym "ML" be withdrawn from the manuscript and replaced by "QRF" to avoid ambiguity.

We agree and have adjusted the manuscript.

(d) Lines 85-87: Would the results change if the validation period was different (i.e., two years in the beginning instead of the end, or even two random years picked in the 10 years). Testing the sensibility to this choice would make the results even more robust.

This has been tested for Halmstad (see answer comment b). Also, additional simulations testing the temporal sensitivity have been done in using 10, 20 or 30 years as different setup periods for all possible x/y sets (table r3). In the manuscript, table 1 has been reviewed accordingly (but we did not want to include the full table r3 for clarity and readability in the paper itself).

(e) Lines 91-92: Based on each x/y predictor-reconstruction station pair, a linear equation is found using the least squares method as means of determining the best fit coefficients.

It could be nice to see the coefficients and try to interpret their meaning : I assume they would be positive (high sea level at one station means high sea-level at another station) and would

reflect both

- the ratio of intensity of sea-level variations between stations

- the correlation between stations.

Even if you choose not to show the values of the coefficients in the final version of the article, I would like to see them as a sanity check, and I believe a little comment on our interpretation of the coefficients would be nice in the manuscript, even if you do not show them.

The table r2 below shows the maximum and minimum Linear fit coefficient values for each simulation across each x/y predictor-reconstruction pair. This following part has been added to the paper: "As expected, all coefficients values are positive and fairly close to one (0.765 to 1.12) meaning low sea level at one station corresponds to low sea level at another station and similar effect is then found for high and intermediate sea level. Therefore, the sea level at one station varies at a rather similar rate to the other one as a coefficient value of 1 would mean that the sea level measured at one station would be increasing or decreasing as the same rate at another one. The set xHornbaek/y_Halmstad presents the closer to one coefficient highlighting a strong correlation between those 2 stations. Only the xRinghals/y_Halmstad and x_Viken/y_Halmstad present a coefficient higher than 1. This suggests that the sea level at Halmstad varies at a higher pace than at the two predictor's stations."

*Table r2. Linear fit coefficients values intervals for each simulation across each x/y predictor-reconstruction pair.*

| Linear Fit coefficients | | | | | | | | |
|---|---|---|---|---|---|---|---|---|
| xHornbaek_ yRinghals | xHornbaek _yViken | xRinghals_y Hornbaek | xRinghals _yViken | xViken_y Hornbaek | xViken_yR inghals | xHornbaek_y Halmstad | xRinghals_y Halmstad | xViken_yH almstad |
| [0.765 ; 0.837] | [0.838 ; 0.873] | [0.853 ; 0.925] | [0.891 ; 0.929] | [0.936 ; 0.981] | [0.907 ; 0.951] | 0.985 | 1.06 | 1.12 |

(f) Lines 92-93: It feels like the predictor for the linear regression and QRF could have been chosen to be a little bit more complex, revealing other time-space relationships between the stations. Since the sea level is sensitive to meteorological conditions, which are advected by the winds, the sea level at one station at time t might be better predicted if using time-lagged sea-level from another station, e.g. the sea-level at times t± a few days (this could be tested very easily from your algorithm, at least for a couple of stations). Even time-delayed embeddings could be used, with sea-level at times t − mdays, ..., t − 1day, t, t + 1day, ..., t + ndays where m and n would have to be optimized. Although this might be out of the scope of the study, it should be tested or at least mentioned. But maybe you have reasons to believe that this would not be

useful/necessary ?

This is an interesting point we have been thinking about. Indeed, as we are looking into predictions from spatially apart stations, physical variables travel resulting in some time lags effects after between 2 spatially distant stations. Here, we are using daily maxima which might buffer this effect to some extent. However, it is most likely some slight improvements could be found when applying some time-delayed variables. Therefore, a short analysis has

been done in testing time lagged variables for the setup period of 30 years for each x/y stations pair. 3 different tests have been done, the first one is the one used in this paper where no time-delayed predictors have been added: $y_t = RF(x_t)$; in the second one we add 2 times delayed variables time t-1day and time t-2days: $y_t = RF(x_t, x_{t-2}, x_{t-1})$; in the third one we add 4 times delayed variables time t-1day, time t-2days, time t+1day and time t+2days: $y_t = RF(x_t, x_{t-2}, x_{t-1}, x_{t+1}, x_{t+2})$. Slight improvements of RMSE values of around 1 to 2.5 cm as well as r values of around 0.03 to 0.08 for all x/y stations pair with best test being the third one and the second one presenting intermediate improvement. Bias values barely changed with changes of maximum 0.2 cm however, towards the extremes, values from the tests 2 and 3 present a bigger underestimation than the ones from the test 1. We then think that, for this study, the method used within the paper (test 1) is sufficient and even might be the best one to reproduce ESLs. This might be explained as the RF is only statistically based and applied on stations not so far away from each other (max 130km) which are therefore most of the time submitted to the same synoptic atmospheric phenomena within one day. Some more testing could be done to really assess the potential added values of using time delayed variables but this is outside of the scope of this study.

(g) I believe section 2.2.2 and section 2.2.4 need to be clarified. Although I am not an expert of QRF, it seems to me that your formulation is misleading.

Lines 96-97 The QRF method yields a mean and a standard deviation for each predicted value (Breiman, 2001; Meinshausen, 2006. You should add equation to this sentence to make it clear. QRF estimates quantiles of a predictand, not average and standard deviation, therefore your formulation is confusing. What do you call "predicted value" ? You need to be specific and to use precise vocabulary. What you should do is to describe the method in detail, to help unfamiliar readers. I suggest to extend this paragraph and to add equations.

Indeed, RF estimates quantiles of a predictand. Here we also estimated the [0.25 0.5 0.75 0.9 0.95 0.97 0.99] quantiles (using the quantilePredict MATLAB function) but we decided not to use them for our analysis both for simplicity and because we did not see clear and systematic improvements between each simulation so, you are right that we are using random forest (RF) rather than QRF and this has been updated in the document. Instead, we used the MATLAB function "predict" which in our implementation uses the stated Treebagger function is used to predict responses using ensemble of bagged decision trees (https://se.mathworks.com/help/stats/treebagger.predict.html). Using this function, we obtain the mean and standard deviation values that we further use within the analysis. We use the predict function for our regression problem and in the documentation there is further description for the  function for the weighted average of the prediction using selected trees. We dont use the option of TreeWeights but do use the outputof the standard deviations of the computed responses over the ensemble of the grown trees, hence for regression: [Yfit,stdevs] = predict(B,X). Here, Yfit is a vector of predicted responses for the predictor data in the table or matrix X, based on the ensemble of bagged decision trees B. By default, predict takes a democratic (nonweighted) average vote from all trees in the ensemble. Given the non-weighted approach we felt additional equations in the manuscript may be confusing more than helping but we provide the reference links to the documentation of the functions used for the interested reader.

 Some further work using the predicted quantile values could be of value but this is, we think, out of the scope of this technical note. We also think applications having larger amounts of data than our will benefit further from

estimates of quantile information and higher order moments but for our case prediction of the mean and standard deviations was decided to be used.

Lines 97-98 The QRF model is implemented using the MATLAB function TreeBagger where the regression method is based on a number of trees and minimum leaf size hyper-parameters. Reading the MATLAB documentation indicates that TreeBagger cannot be used alone, another function must be used to perform the regression. Did you use "quantilePredict" ? "predict" ? "fitrensemble" ? Something else ? Please specify this to allow reproducibility and help understanding.

See previous comment, this has been added in the paper using the function "predict".

Line 98 These parameters are here set to 500 and 1, respectively. Consulting the documentation on TreeBagger MATLAB method, it seems that 1 is the default parameter for classification trees. Some justification for the choice of 500 would be nice here.

By default, the MinLeafSize parameter for regression models is 5 (and 1 for classification). These hyperparameters have been chosen after a brief sensitivity analysis, where MinLeafSize chosen due to best results, especially towards the extremes and 500 is chosen to ensure convergence of the RF model among the testing sites. This has been made clear in the manuscript.

Line 219 QRF method with random sampling to evaluate return levels (RLs) "QRF method with random sampling" is not a known terminology, it is one you designed for the purpose of this study. Therefore, it must be made very clear in the manuscript. For instance, you could say "in the following, we denote "QRF method with random sampling" the following methodology:..." and then describe your methodology. The description must be very clear and thorough, including

every step of the calculation, to avoid misunderstanding and allow reproductibility of your results. Also, since this method is compared with another way of estimating return levels (Figure 4) you should explain this other method of estimating RLs (simply named "QRF" in Figure 4) in this section as well.

This has been added in section 2.2.4.

Lines 120-122 Based on the QRF daily means and standard deviations, we calculate the corresponding annual maxima from the reproduced time series and their associated standard deviations. This isn't clear to me. I suppose "the corresponding annual maxima" are the annual maxima of average QRF predictions ? But how do you incorporate the standard deviations in the maxima ? You should write equations for this.

This has been added in section 2.2.4.

Lines 122-123 We assume that a Gaussian distribution describes the probability for each predicted annual maximum. It seems that this is a hypothesis that you could (and should) test for.

This is an arbitrary choice introduced with the RF method with random sampling that, you are right, might not be the best one in terms of distribution fit. However, in the context of this technical note, we think it is enough to introduce the method in this way. In general, more data is needed to make use of other choices of distributions (with for instance introduced skewness or shape parameters) compared to the simple Gaussian assumption. For this reason we decided to maintain this choice for this paper.

Line 123 10 000 sets of 30-year maxima You have Gaussian distributions for annual maxima, and you use it to draw 30-year maxima ? Why 30-year maxima ? What does this mean ?

The revision of the section 2.2.4 should clarify this point.

Line 125 This yields an ensemble of randomly drawn RL curves. Why would you trust this method rather than simply using the QRF-mean daily maxima ? If one method is better from a statistical viewpoint, then there is no point in doing both (and showing both in the report). This adds confusion. I suggest you consider keeping only one, either "QRF" or "QRF random sampling". If not, this choice should be motivated. Due to this confusions, Figure 4 appears unclear to me, while it is the most important figure of the pre-print. Maybe it is due to my lack of knowledge of QRF methods, but I doubt that this is the only reason. Anyway, this technical note should be accessible for readers unfamiliar with QRF.

You are right and, we think, your comment relates to our miscommunication and clarity with the proposed new approach which, we hope, has been solved with the revision of section 2.2.4. We would rather keep figure 4 as it is as it permits to see the difference between both approaches and highlights the added value of the new method "RF with random sampling".

(h) Figure 3 Since you show only one example, I think it would be better to show one with Halmstad as predictand, as this is the main objective of your study. This would also allow you to illustrate the points mentioned here in comment (a).

We would rather like to keep a more representative figure of how the method is applied, especially towards the extremes. Presenting one from Halmstad does not allow us to test the model over a long time period (only 2 years for Halmstad) and therefore loose information of checking the method for ESLs. The idea of checking the method for all the different sets is especially done to understand how it behaves on long time series and towards the extremes. And, therefore give us confidence when we apply the same method to predict Halmstad sea level based on a neighbor station.

We re-ran the analysis for all temporal and spatial combinations possible with data being first linearly detrended which changed the numbers a little. Indeed, we first did not detrend it because we were dealing with only short period lengths of data but now that we also analyzed on longer time series (20 and 30 years), we thought it is better to actually detrend it. The conclusions did not change.

**3   Technical corrections**

1. I think using only "ESL" and not "ESLs" would be enough, and clearer. However this decision is yours to make.

To keep the consistency with the plural used as when mentioning return levels: RLs, we left (or deleted) the "s" or not accordingly.

2. Figure 1: Some of the fonts are too small to be read (the latitudes/longitudes, as well as the city names on the left panel). Either enlarge the font or supress the text.

We agree and have adjusted the manuscript.

3. Line 71: from which the annual (daily) maximum → you could remove "(daily)".

We agree and have adjusted the manuscript.

4. Lines 76-77: Conversely, long-term linear trends (i.e., sea level rise) were estimated for all time series and found to range between 0.34 and 1.47 cm per decade.

Could you indicate all values of computed linear trends, along with the corresponding city ? Since there are only 4 stations it would not be excessively lengthy. Also, I think the way these linear trends are estimated should be

explained in a bit more details. There are different ways of estimating linear trends for sea-level, corresponding to different hypothesis. In particular, for the Hornbaek station, is the linear trend computed using the whole time series (i.e. before 1900) ? Is this relevant or should the rise start later ? Does it make any difference for the estimated time-series ? Although it might not make a huge difference, it seems important to indicate this, since you are estimating long return periods and since Hornbaek is the longest time-series in your dataset, and therefore the time-series which contains a large part of the information on which your ML techniques rely. "It is worth noting that since we use observed tide gauge data, long-term trends, that is, climate change induced sea level rise are implicitly considered, although site-specific changes in the relative sea levels due to, e.g., human activities may introduce biases." Same here. These points are crucial to your study and need to be debated more.

We added the trend of all time series in the text and the approach used (linear trend calculated over the whole time series length). We also checked if the rise happens later for Hornbaek in looking at the decadal trend but there were no clear signs of such effect when looking at decadal trend values.

5. Lines 83-84: The proposed approach for extending short sea level time series uses one neighbouring station as predictor (station x) of past sea level data at the station of interest (station y). The way x and y are defined is not fully clear. I assume that you are using daily maxima, with long-term linear trend removed. Please recall this here. We rephrased it to hopefully clarify it.

6. Lines 92-93: the sea level at station y is predicted from the sea level at station x. Perhaps you should make it clearer that it is the sea level at station y, time t, that is predicted from the sea level at station x, time t. See comment above for the same lines 92-93.

We agree and added this information on temporal resolution.

7. Table 3: You provide "uncertainties" from the paper by Andersson (2001), however, I cannot find an explanation of what these uncertainties are, more precisely. This would help to compare it with your "95th percentile ensemble spread".

According to Andersson (2021), the uncertainties found are the 95% confidence interval resulted from the best suited distribution fitted on yearly maximum values extracted from the time period from July to June. Hence, a similar approach as is standard in extreme value analysis (GEV) on observed data was used based on what they refer to as yearly maximum values representative of each location.

8. Also in Table 3: I think you should be able to give uncertainties associated with the QRF-based RLs from every station, and therefore add a line of the type "uncertainties" below each line. This is a type of output available from a QRF model I believe.

The uncertainties associated with the RF-based RLs from every station are visible in Figure 4 and therefore, to keep the paper as a technical note and not extend it too much, we think it is enough to not show the uncertainties related to this method but really focus on the proposed method we named "RF with random sampling" method.

9. Also in Table 3: Are the RLs computed by Anderson based only on Viken as predictor, or are the winds also used as mentioned in the text ? If this is the case, you should specify it in the table with something like "Viken + wind". If not, you should specify it in the text to avoid confusion. Also, I think a bit more description of Andersson's method would be helpful here, since you use it many times for comparison.

Added "Viken+wind" as proposed. we added the little information we had about this method.

It is difficult to find more information about the exact method used here, a reference from the cited paper orientate us towards this following reference ("Johansson, L., (2018) Extremvattenstånd i Halmstad."). From the cited source, we can read: "For Halmstad, no suitable long time series of observed sea levels is available. However, the Swedish Maritime Administration (SMA) has performed measurements of sea levels in Halmstad from 2009 and onwards. This time series is too short for direct use in the present analysis. Estimates of return values with a return period of up to 200 years would not be reliable. Therefore, sea level observations from Viken station in combination with wind speed observations from Nidingen station have been used to obtain a longer data series describing conditions representative for Halmstad. For more details on observations included in the analysis, see Johansson, 2018." It is however not clear exactly how they calculate the 'wind part' when reading it carefully (in Swedish). They do state that it is better to use the wind as an average for some hours before the event and that they somehow take into account of the wind turning also during this time but without providing a clear algorithm. Nevertheless, they extended a time series for yearly maximas for 36 years from 1982 to 2017 and applied a lognormal distribution fit and provided the 95% confidence interval of this fit.

10. Lines 193-194: "Here, observed values are added to the extended time series to get the longest time series possible before a GEV fit is applied." You should be more specific. Which observations are added ? How many years/months ? How much does that strengthen your model ?

See answer comment 33. No specific testing has been done to check on how much it strengthen the model but we believe that using observations when available for real applications will yield to better results as observations are not associated with uncertainties and are seen as the "true" values (as they are "only" limited to the instrument used for measurement).

We accordingly changed the sentence a bit to add some precision about it "Here, the full-period length of Halmstad's observed values (station y) are concatenated with the predicted time series to get the longest and more accurate extended time series possible before a GEV fit is applied."

11. Lines 194-195: "Even so, RLs are still lower than the ones displayed by Andersson (2001) when based on the ML mean outputs (fig. 4-a; Table 3)." How could you explain this systematic bias ? What does it reflect ? Also, I think you should mention that the estimated RLs in Table 3 are all in the uncertainty range of Andersson's study (2001), which is a good sign, except for the 200-year RL with Viken as predictor.

Added line 194. "even though within the uncertainty range" and "This can possibly be explained by the underestimation found towards the extremes on the predicted and therefore extended time series. This is why we introduced the RF method with random sampling which permits to capture more extremes values." By this we mean that we can represent extreme values out-of-sample with the method, but only in a statistical sense and with limitations as discussed previously that other choices of distribution functions could potentially be chosen to represent the extreme tail if sufficient data would be available to evaluate the differences.

12. Line 130 RMSE RMSE is not shown → perhaps add it to the table, in [cm]. It would give an idea of the relative importance of the biases, which is not clear here: a few centimeters seems to be small, but if we don't know the amplitude of typical variations of sea-level there is no way to really know (and this information is station-dependant). OR you could show the relative biases in the Table (for instance: baises normalized by RMS sea-level decadal variations around the mean)

Instead of r values, RMSE are shown in the table 2 in the manuscript (to not overload the table and keep clarity). As, in Figure 2 and section 2.1, the sea level time series are presented along with an added small description, we do not think it is necessary to introduce normalized values.

13. Line 131 perc95-bias I do not see this in Table 2 ?

Right, changed to "not shown". Also, line 129, we changed "(table 2)" "(to partly presented in Table 2)". We decided to not expand the paper too much to increase clarity and readability.

14. Lines 131-133 For the annual maxima, the 95 th , 97 th , and 99 th percentiles sets, marginally higher r and lower RMSE values are found for the LR in nearly all cases, with a maximum difference of 6 cm for the RMSE and 0.10 for the r value Not shown. Also, how can that be understood together with the fact that the biases are somewhat smaller when using the QRF ? It seems counterintuitive, this should be explained.

As RF can account for possible non-linearity effects, this can explain, we think, the smaller biases found, especially towards extremes. The higher r and lower RMSE found for the LR might be explained because of the data being slightly more scattered using the RF. Updates of the values with the new analyses done and more details have been given.

15. Lines 133-134 Overall, RMSE values are between 10 and 40 cm, and r values are between 0.6 and 0.9 in most cases. what does that indicate ?

In comparison between the extreme sets and the full time series, we can see that: "This highlights the fact that both models lose accuracy to predict ESLs compared to predicting less extremes events. And this seems to be caused by the non-linear effects occurring during the extremes as the decrease of r shows." (added in the text)

16. Line 135 a slight underestimation of the observed extreme values for both the LR and QRF how do we know that -30cm is "slight" ?

As mentioned earlier, some comments have been added in section 2.1 when presenting the observation time series as well as the figure 2. We think this is enough to understand the range and behavior of the observation time series.

17. Line 138 is observed in nearly all cases add "not shown".

We agree and have adjusted the manuscript.

18. Lines 140-144 In those two cases, the QRF does not correctly reproduce the extreme range, as they are out-of-sample while the predicted values are bounded, since the ML can only reproduce in-sample events. Compared to an LR, it is clear that the inherently non-linear QRF is better able to account for the few moderate extremes that occur during the 8-year training period, whereas they are likely to be suppressed in a linearized model. To me, this is a very interesting point here. You seem to conclude that the QRF is better than the LR, since the latter smooths out the extremes, however you also point out that the QRF is not able to produce out-of-sample predictions. I would recommend a more nuanced conclusion.

As we are mentioning the "moderate extremes" (because of the short length of the time series), this holds. We can also add that, even when the RF is having trouble with reproducing extremes, it seems it is not worst / more underestimating those than the LR in most cases (check new results). A few sentences with more explanation have been added towards this point.

19. Figure 3 - Standard deviation is also available when performing LR (it is assumed to be always the same, this is called homoscedasticity), it is given by the RMSE, therefore you should also plot the error bars for the LR. - It

seems like you are not using QRF but simply RF, since you estimate a mean and standard deviation, am I mistaken ? - I see coloured stars close to the diagonal y=x but I do not see how these could be 1st and 99th percentiles ? This part isn't clear. - The figure is quite fuzzy in the dense area of average sea levels between 0 and 50cm. I recommend that you do not show all error bars, perhaps only for extremes (i.e., above/below certain quantiles) as this is the main objective of your study.

We agree with you and added the standard deviation on the LR plot too. We actually did use a QRF and run simulations for different quantiles but we decided to only focus on the mean and std (RF) for simplicity. Indeed, it looks a bit fuzzy however We would rather like to keep it that way as we are afraid to lose in clarification if only showing std for extremes (as all values can be associated with a std).

20. Line 152 the model accuracy clearly decreases It does not seem "clear" to me. For instance, the best r values (0.91, LR) are found for the pair Viken-Ringhals and Ringhlas-Viken, which seem to be pretty far apart on the map. Only a more systematic study of the statistics (r, biases, etc.) with respect to the distance between the cities would reveal undoubtedly this distance-dependency (which is probably true).

Yes, this is a miscommunication from my side. We analyzed statistics between each x/y pair for each setup period and a more nuanced conclusion has been stated. Also, this decrease is mainly observed for the extremes which we added in the text. This also goes back to the answer from comment 23. We were also looking at the sets towards the extremes to draw this statement.

21. Line 153 around 0.7 I don't see this value in Table 2 ?

See answer comment 23.

22. Line 153 (9 km apart) perhaps show distances somewhere in the Table to improve readability ?

We agree and have adjusted the manuscript.

23. Line 153 r coefficients around 0.3 I don't see the value "0.3" in the table ?

A sentence has been added to explain where this number comes from "when looking at values on daily maxima but also on the extremes sets," (line 152-153). This number comes when taking into account the correlation coefficients from all sets (daily, annual maxima, 95th, 97th and 99th percentiles sets). Some figures could be added as supplementary material on demand to illustrate where all numbers come from.

24. Lines 155-156 (annual maxima or 95 th, 97 th and 99 th percentiles). add the mention "not shown"

Added on next sentence (comment 25)

25. Lines 156-157 similar results are found when comparing the sea level time series for Hornbaek and Ringhals, based on Viken data, and when comparing predictions for Viken and Hornbaek sea levels based on Ringhals data. add "not shown"

We agree and have adjusted the manuscript.

26. Lines 159-160 This can probably be explained on physical grounds however, this is beyond the current technical note. There is nothing we can do with this information, I think it should not be specified, or you should say more.

This is a discussion point to start understanding why mirrored sets do not always present similar statistics, I changed it as follow: "This can probably be explained on physical grounds due to localized phenomena resulting for

example from the topography or the local meteorological conditions, however, this is beyond the current technical note.".

27. Lines 160-161 In general, the QRF method seems to be more accurate than the LR when predicting local sea levels from stations located further away e.g., between Ringhals and Viken / Hornbaek as compared to Viken and Hornbaek Should we see this in the annual maxima ? It is not true when we look at the predictions of Ringhals based on Viken, or Hornbaek based on Viken (2000-2010, 5cm stronger bias for the annual maxima using QRF). **More generally, you should always mention which numbers in the Table support your claim otherwise it can be questioned.**

You are right and this is not so clear and this has been deleted as it would need further investigation to assure this conclusion.

28. Table 2 About the colors, is it the right choice ? You do not indicate why the sign is so important that you highlight it in colour. Red highlighting connotes "danger", should we fear an overestimation of sea-level ?

Colors have been deleted from the table 2 to be in agreement with EGU journal policy.

29. Table 2 You highlight in bold when QRF is better than LR in bias on annual maxima by 5cm. To be fair, you should also highlight (differently) when LR is better than QRF by 5cm, this is the case for the couple Viken-Hornbaek 2000-2010.

We agree and have adjusted the manuscript.

30. Line 183 the setup period replace by "the train and validation period"

We decided to replace "the setup period" by "the training period" only as, in the sentence, it induces that the validation / testing period is then really short.

31. Line 184-185 When analysing the error metrics over the testing period, the model based on Viken station presents the best results. What makes you say that ? The biases are smaller with the Ringhals station as predictor.

Predictions from Viken results in a lower RMSE (7.07 cm against 8.35 cm Ringhals & 11.28 cm Hoonbaek), higher r (0.95 against 0.93 and 0.88 respectively), similar bias (0.91 cm against 0.26 cm and 1.0 cm) and better, slight overestimation perc95-Bias (1.4 cm against -1.5 cm and 2.1 cm). Also, visually, a slight overestimation is visible which makes me think it might be a better fit overall if we had a longer time series to train the model on because of the testing between the different sites made previously. This underestimation is also visible when comparing results between the LR and RF (LR underestimating slightly more the extremes).

32. Line 189 (not shown However this is probably the most important piece of information !

This has been changed in agreement with one of the previous comment where more analyses using different testing periods have been done.

33. Lines 193-194 Here, observed values are added to the extended time series to get the longest time series possible before a GEV fit is applied. What are these added values ? You have to be more specific for readers to know what you have done.

Replaced the word "added" by "concatenated" and added "and more accurate" in the sentence. We have 2 time series possible, the predicted one and the observations one, what we did is to use the full-length observations time series and extend this observations time series using the predicted one (concatenation of both time series).

34. Table 3 - Station x you could add "predictor". - 5th line, for the Andersson (2021) study, you write simply "Viken", but I understand that winds are also used to make this estimate ? Therefore you should perhaps write "Viken+winds" in the first column, 5th line of the Table.

We agree and have adjusted the manuscript.

35. Line 203-204 The inferred RLs are slightly higher than the RLs derived directly from observations are these observation-based RLs shown anywhere in the paper ?

Yes, the observation-based RLs are in blue in the Figure 4.

36. Figure 4's caption (end) Black error bars show RLs and 95 th percentile CI calculated from Andersson (2021). → this should also be in the legend on the right.

- also, what is "MLE" in the legend ?

- also, all elements of the legend should be in the same box, here it seems like the upper box is for the upper panel and the lower box for the lower panel, which is not the case

We agree and have adjusted the manuscript. "MLE" stands for "Maximum Likelihood Estimation".

37. Lines 232-233 This limitation is a known issue when applying ML-based prediction models (Tyralis et al., 2019; Hengl et al., 2018); Wrong use of "ML" : many machine learning algorithm can produce values outside of the observed range. The two cited paper are about random forests, not ML in general.

Replaced "ML" by "random forests"

38. Line 239 but it may also confuse the interpretations at times, could you be more specific ?

More complex methods often involve a large number of parameters or employ advanced algorithms such as deep learning techniques. They can capture subtle patterns that simpler models do not. However, because of their complexity, such methods can become harder to understand exactly how the model is making its predictions and therefore to interpret. This is particularly true for neural networks for example, acting as "black-box" models where the learnt relationships can be highly abstract.

39. Line 241 but this is certainly an active research area I think you should replace "active" by "promising"

We agree and have adjusted the manuscript.

40. Line 250 The best reconstructions are generally achieved for stations spatially closer maybe this would change if you allow to use time-delays in the definition of x. See comment (f) above.

See answer comment (f).

41. Line 252 We tested the QRF method with random sampling Replace by "We tested another method that we named 'QRF with random sapling'."

We agree and have adjusted the manuscript.

42. Line 281 That doi seems to point to another article which is not the one you mention in the text.

We agree and have adjusted the manuscript.

*Table r3. Experimental setup and summary of analyses.*

| Predictor | Predictand | Setup period | | | Coinciding period | Distance between stations | Study |
|---|---|---|---|---|---|---|---|
| station x | station y | 1 | 2 | 3 | | (km) | |
| Hornbaek | Viken | 2010-2020 | | | 1977-2020 | 9 | |
| | | | 2000-2010 | | | | |
| | | | | 1990-2000 | | | |
| | | 2000 - 2020 | | | | | |
| | | | 1990 - 2010 | | | | |
| | | 1990 - 2020 | | | | | |
| Ringhals | Viken | 2010-2020 | | | 1977-2020 | 127 | |
| | | | 2000-2010 | | | | |
| | | | | 1990-2000 | | | |
| | | 2000 - 2020 | | | | | |
| | | | 1990 - 2010 | | | | |
| | | 1990 - 2020 | | | | | |
| Hornbaek | Ringhals | 2010-2020 | | | 1968-2020 | 130 | |
| | | | 2000-2010 | | | | |
| | | | | 1990-2000 | | | |
| | | 2000 - 2020 | | | | | Spatial and Temporal correlation analysis |
| | | | 1990 - 2010 | | | | |
| | | 1990 - 2020 | | | | | |
| Viken | Ringhals | 2010-2020 | | | 1977-2020 | 127 | |
| | | | 2000-2010 | | | | |
| | | | | 1990-2000 | | | |
| | | 2000 - 2020 | | | | | |
| | | | 1990 - 2010 | | | | |
| | | 1990 - 2020 | | | | | |
| Ringhals | Hornbaek | 2010-2020 | | | 1968-2020 | 130 | |
| | | | 2000-2010 | | | | |
| | | | | 1990-2000 | | | |
| | | 2000 - 2020 | | | | | |
| | | | 1990 - 2010 | | | | |
| | | 1990 - 2020 | | | | | |
| Viken | Hornbaek | 2010-2020 | | | 1977-2020 | 9 | |
| | | | 2000-2010 | | | | |
| | | | | 1990-2000 | | | |
| | | 2000 - 2020 | | | | | |
| | | | 1990 - 2010 | | | | |
| | | 1990 - 2020 | | | | | |
| | | *Setup period* | *Testing period* | | *CASE STUDY* | | |
| *Hornbaek* | *Halmstad* | *2010-2020* | *2010-2012* | | *2010-2020* | *68* | *case study* |

| | | | 2011-2013 | | |
|---|---|---|---|---|---|
| | | | 2012-2014 | | |
| | | | 2013-2015 | | |
| | | | 2014-2016 | | |
| | | | 2015-2017 | | |
| | | | 2016-2018 | | |
| | | | 2017-2019 | | |
| | | | 2018-2020 | | |
| *Viken* | | *2010-2020* | 2010-2012 | *60* | |
| | | | 2011-2013 | | |
| | | | 2012-2014 | | |
| | | | 2013-2015 | | |
| | | | 2014-2016 | | |
| | | | 2015-2017 | | |
| | | | 2016-2018 | | |
| | | | 2017-2019 | | |
| | | | 2018-2020 | | |
| *Ringhals* | | *2010-2020* | 2010-2012 | *80* | |
| | | | 2011-2013 | | |
| | | | 2012-2014 | | |
| | | | 2013-2015 | | |
| | | | 2014-2016 | | |
| | | | 2015-2017 | | |
| | | | 2016-2018 | | |
| | | | 2017-2019 | | |
| | | | 2018-2020 | | |

---

## Author Comment (AC2)

*REVIEWER1: General Remarks on the Article:*

1. *The article is well formed and works on a significant issue. As there are so many water level stations throughout the world. Many stations have missing data or long gaps. The proposed method can help fill these gaps, especially with neighboring water level stations.*

Thank you for your review and general positive comments about our study. We gave our responses below marked in red.

1. *One general negative comment is that, while talking about extreme sea levels, the authors do not talk about the storm surge or similar phenomena. Or in general if the authors are dealing with which extreme sea level events.*

Lines 39. / 40. Hypothesis tides are not large enough in the areas and therefore we did not take them into account. (storm surges and extreme sea levels are assumed to be almost the same water levels). We added a reference to Svansson, 1975. We initially did study the tides on the West Coast of Sweden, but found them to be small in the Southern part which we focus on here.

1. *I would suggest a change in the title of the manuscript. The current title suggests that the main focus is going to be about Machine Learning. However, when the overall manuscript is considered, it feels more statistical (as per the topic) than the ML part.*

We agree and have adjusted the manuscript with the new following title: "Extending sea level time series for extremes analysis with statistical methods and neighbouring station data".

1. *As mentioned below, I believe the geographical location of the stations are very important. Hornbeck and Viken stations are constricted in a channel. In a tidal setting this will change how the water level behaves. This might be a big difference even in the characteristics of the water level time series. I believe this should be mentioned in the manuscript (event if it is not considered in the analysis).*

You are right and the following sentence has been added in the section 2.1: "The geographical location of the stations is important as it can change how the water level behaves, for example, if the stations are constricted in a channel as for Viken and Hornbaek. Here, ESL are defined as the total highest measured sea level including tides and storm surges, this choice is motivated because of the low tidal range in the area (Svansson, 1975).".

1. *Between L39-50 authors mention many different methods and analysis. It would have been quite good to mention, how good the presented method compared to some of these studies.*

It is difficult to say as this would need to compare those presented methods between each other on a systematic framework which is quite extensive work to achieve. Each method would also need to be described which would make such a manuscript significantly longer. We think that this is outside of the scope of this technical note but could bring a great value to another study.

*Small Remarks on the Article:*

1. *In the abstract there are many vague words, that has to do with the definition of the time series or quality of the outcome. For example, "Reasonable" is one of them. It would have been better to define the quantity and statistical measure.*

To keep the abstract short, I did not want to go to deep and define the quantity and statistical measures. But we agree with your comment to be clearer and more precise about things and in the revised manuscript we hope to have clarified a number of issues brought up by especially reviewer 2.

1. *Between L30-40 there is a small definition of the data time series. Although the length is defined, there is no indication of the interval of the data until the section 2.1. It would be better to define the interval of the data, since it will also provide insight on the number of data points.*

I understand your comment but would rather keep it this way as the introduction is, I think, introducing the field and general background behind the study. The data are then introduced in more detail in section 2.1

1. *Also in the same part, the highest record of 235 cm is given. It would have been a good idea to explain the event, as mentioned in the previous comment. Is it a storm surge or happened during spring tide etc.?*

It is difficult to understand exactly what happened for this particular event and therefore to introduce it in the paper, I think. According to Johansson, 2018, this event was mainly due to local conditions leading to a sea level increase of 50 to 100 cm in comparison with neighbouring stations as Viken, the second one is a seiche effect which could add around 25 cm to the total sea level. We have now briefly described this in the manuscript and referred to the reference (Johansson 2018) that studied this extreme event.

1. *In the methods part it is not clear which data is used for LR for QRF methods. Is it the hourly data or the daily data? In case if it is the hourly data, how good a good a fit is obtained using LR method to a tidally harmonic data?*

As mention in section 2.1, the daily data are used throughout the full analyses and precision has been added in this section 2.2.

1. *In L100 the sentence says the LR model is trained, but since it is a Least Squares Method, I don't thing "trained" is the correct word. It would be better to say "the LR model is fitted".*

We agree and have adjusted the manuscript.

1. *If Figure 3 is showing the Setup Period 1 (as far as understood, it should be noted in the caption).*

We agree and have adjusted the manuscript.

1. *In Section 3.1 one of the metrics is RMSE. Although it is a good metric, for example 6 cm RMSE in a 200 cm water level vs 30 cm water level is quite different. I suggest to use either a normalized RMSE or giving the range of the water level within Table 2.*

On suggestion from reviewer 2, we chose to not normalize it but introduced a bit more details about the time series for each station in section 2.1. Also, the figure 2 permits, we think, to get an overall understanding of the time series behaviour.

1. *In general, and discussed in between L150-160, there are two sets of stations with very significant geographical differences. Hornbeck and Viken stations lie inside of a channel (almost at the entrance). However, compared to these stations, other stations are on the open coast. Maybe this is what's meant in L159-160 by the physical grounds, but this might be a big difference even in the characteristics of the water level time series.*

This has been added to the paper, thank you.

1. *L194 and Table3 Andersson is given two different dates (2001, 2021).*

We agree and have adjusted the manuscript.

1. *In Figure 4 the colors of the stations are over washed by the shadow colors. Different color scheme or changing the line properties might help.*

We agree and changed the line properties, it is however quite difficult to find a good color scheme. We are happy to continue to work more on these technical aspects of improving the quality of figures if needed during the later stages of the review process.

---

## Referee Report (RR1)

**2nd (and final) comments on "Technical note: Extending sea level time series for extremes analysis with machine learning and neighbouring station data"**

Anonymous referee #2

November 6, 2023

Note: quotes from the pre-print are in blue.

**1 General comments**

The authors have made significant changes to the manuscript which is now much clearer and more precise. Also, several supplementary numerical tests have been performed, which strengthen the analysis and make the results more robust. I thank the authors for their dedicated work and for making a very clear answer to all the comments.

For these reasons, I suggest an acceptance of the paper as is. Two typos are listed below.

**2 Typos**

1. Line 93 (x)constituting missing one blank space here

2. Figure 3's caption Blue points show the daily maxima and each corresponding blue bar shows the standard deviation (b) I think that this "(b)" is an error.

---

## Author Response (AR2)

2nd (and final) comments on "Technical note: Extending sea level
time series for extremes analysis with machine learning and
neighbouring station data"

Anonymous referee #2

November 6, 2023

Note: quotes from the pre-print are in blue.

**1 General comments**

The authors have made significant changes to the manuscript which is now much clearer and more precise. Also, several supplementary numerical tests have been performed, which strengthen the analysis and make the results more robust. I thank the authors for their dedicated work and for making a very clear answer to all the comments. For these reasons, I suggest an acceptance of the paper as is. Two typos are listed below.

Thank you for your review and your positive answer. Our responses are given below marked in red.

**2 Typos**

1. Line 93 (x)constituting missing one blank space here

We agree and have adjusted the manuscript, thank you for spotting it.

2. Figure 3's caption Blue points show the daily maxima and each corresponding blue bar shows the standard deviation (b) I think that this "(b)" is an error.

We agree and have adjusted the manuscript.